# No-Regret Learning in Strongly Monotone Games Converges to a Nash Equilibrium

## Abstract

This paper studies a class of online games involving multiple agents with continuous actions that aim to minimize their local loss functions. An open question in the study of online games is whether no-regret learning for such agents leads to the Nash equilibrium. We address this question by providing a sufficient condition for strongly monotone games that guarantees Nash equilibrium convergence in a time average sense, regardless of the specific learning algorithm, assuming only that it is no-regret. Furthermore, we show that the class of games for which no-regret learning leads to a Nash equilibrium can be expanded if some further information on the learning algorithm is known. Specifically, we provide relaxed sufficient conditions for first-order and zeroth-order gradient descent algorithms as well as for best response algorithms in which agents choose actions that best respond to other agents' actions during the last episode. We analyze the convergence rate for these algorithms and present numerical experiments on three economic market problems to illustrate their performance.

## 1 Introduction

Online convex optimization (Hazan et al., 2016; Shalev-Shwartz et al., 2012) is used to solve decision-making problems where the cost function is unknown and optimal actions are selected with only incomplete information. Recently, online convex optimization has also been employed for the solution of games involving multiple agents with applications ranging from traffic routing (Sessa et al., 2019) to economic market optimization (Narang et al., 2022; Wang et al., 2022; Lin et al., 2020). In these online convex games, agents simultaneously take actions to minimize their loss functions, which depend on the other agents' actions.

Generally, agents in non-cooperative games have access to limited information. For example, they may not be able to observe the actions of other agents and may not even know the exact game mechanism. As a result, rational agents will focus on sequentially learning their individual optimal actions at the expense of other agents, and their ability to do so efficiently can be quantified using notions of regret that captures the cumulative loss of the learned online actions compared to the best actions in hindsight. An algorithm is said to achieve no-regret learning if the regret of the sequence of online actions generated by this algorithm is sub-linear in the total numbers of episodes $T$. While no-regret learning has been studied for a variety of games; see e.g., Sessa et al. (2019); Tatarenko & Kamgarpour (2018); Wang et al. (2022); Daskalakis et al. (2021); Anagnostides et al. (2022), the analysis of regret alone is not sufficient to characterize the limit points of a learning algorithm, i.e., the sequence of actions taken by the algorithm. In fact, no-regret learning may not converge at all and can exhibit limit cycles, as shown in Mertikopoulos et al. (2018).

In this paper, we adopt the notion of a Nash equilibrium, that describes a stable point at which the agents have no incentives to change their actions, to analyze the convergence properties of no-regret algorithms for online convex games. A growing literature has recently focused on showing Nash equilibrium convergence for online games; see, e.g., Bravo et al. (2018); Tatarenko & Kamgarpour (2020); Drusvyatskiy & Ratliff (2021); Lin et al. (2021; 2020); Mertikopoulos & Zhou (2019); Narang et al. (2022); Heliou et al. (2017); Golowich et al. (2020); Azizian et al. (2020). For example, for potential games with finite actions, Heliou et al. (2017) show that the sequence of play returned by the exponential weight algorithm converges to the Nash equilibrium. On the other hand, for games with continuous actions, strong monotonicity, which ensures the uniqueness of the Nash

equilibrium (Rosen, 1965), is a sufficient condition for Nash equilibrium convergence for many specific learning algorithms, including the mirror descent algorithm (Bravo et al., 2018; Lin et al., 2021), the dual averaging algorithm (Mertikopoulos & Zhou, 2019) and the derivative-free algorithm (Drusvyatskiy & Ratliff, 2021; Narang et al., 2022). Besides, an optimistic gradient algorithm is proposed in Golowich et al. (2020) that achieves tight last-iterate convergence for smooth monotone games. Similarly, Lin et al. (2020) investigate the last-iterate convergence for continuous games with unconstrained action sets that satisfy a so-called 'cocoercive' condition that includes a broader class of games with potentially many Nash equilibria. However, all these works analyze the convergence and/or regret for specific learning algorithms and for assumptions that depend on this specific choice of algorithms and games.

In this paper, we follow a different approach and instead focus on understanding for what classes of games and learning algorithms Nash equilibrium convergence can be guaranteed. Specifically, we are interested in understanding whether and for what class of online convex games with continuous action sets no-regret learning converges to a Nash equilibrium regardless of the specific algorithm. Moreover, we are interested in understanding whether and how this class of online convex games can be expanded when the no-regret learning algorithm is known.

In our main result, we show that for $m$-strongly monotone games with parameter $m > 2L\sqrt{N-1}$, where $L$ is a Lipschitz constant of the gradient function with respect to the actions of other agents, and $N$ is the number of agents, any no-regret algorithm leads to Nash equilibrium convergence. While Nash equilibrium convergence has been analyzed for different combinations of learning algorithms and games, to the best of our knowledge, this is the first effort to understand for what classes of games and learning algorithms Nash equilibrium convergence can be guaranteed, and thus bridge regret analysis with Nash equilibrium convergence in games. We note that this result applies to any no-regret algorithm and thus can provide theoretical support for the convergence of any such algorithm for which regret analysis is easy but Nash equilibrium convergence is difficult to show. Furthermore, we show that the class of games $m > 2L\sqrt{N-1}$ can be expanded if additional information about a specific no-regret algorithm is known. First, for the class of gradient-descent (GD) algorithms including first-order and zeroth-order algorithms, we show that $m > 0$ is a sufficient condition for Nash equilibrium convergence. Note that Drusvyatskiy & Ratliff (2021) also show convergence of the zeroth-order algorithm to a Nash equilibrium, but with the additional assumption that the Jacobian of the gradient function is Lipschitz continuous which they need to ensure that the smoothed game induced by the zeroth-order oracle is strongly monotone. In this work, we show that this assumption is not necessary and Nash equilibrium convergence can still be guaranteed even when the smoothed game is not strongly monotone. In addition, we study the class of the best response algorithms, where every agent selects the best action in the next episode given the other agents' current actions. Best response algorithms have been studied for several classes of games, including potential games (Swenson et al., 2018; Durand & Gaujal, 2016) and zero-sum games (Leslie et al., 2020). However, none of these works study games with continuous actions or provides sufficient conditions that guarantee convergence to a Nash equilibrium. We show that for $m$-strongly monotone games, the best response algorithm ensures Nash equilibrium convergence if $m > L\sqrt{N-1}$. This is, to the best of our knowledge, the first convergence analysis of the best response algorithm in continuous games.

We numerically validate the proposed algorithms using three online marketing examples, specifically the Cournot game, the Kelly auction, and the Retailer pricing competition, that satisfy different conditions on the parameter $m$ and, therefore, belong to different game classes. We show that for games that do not satisfy the sufficient condition $m > L\sqrt{N-1}$, such as the Cournot game, the best response algorithm may diverge. As a result, gradient descent algorithms may be better suited to solve games in this class. We also compare the performance of these algorithms for games for which Nash equilibrium convergence is guaranteed. We observe that when $m > L\sqrt{N-1}$, the best response algorithm outperforms first-order gradient descent which, in turns, outperforms the zeroth-order method. On the other hand, when $0 < m < L\sqrt{N-1}$, first-order gradient descent outperforms the zeroth-order method. In summary, by defining sufficient conditions for Nash equilibrium convergence that depend only on the properties of the game, i.e., the parameter $m$, and not the learning algorithm used to solve it, our analysis allows to identify classes of games for which no-regret learning guarantees convergence to a Nash equilibrium without analyzing specific algorithms or identify specific no-regret learning algorithms with no guaranteed convergence to a Nash equilibrium, both fundamental questions in the analysis of online convex games.

The rest of the paper is organized as follows. In Section 2, we define the online convex games under consideration and provide some assumptions. The main result is presented in Section 3, where we provide sufficient conditions on the class of games for which no-regret learning leads to Nash equilibrium convergence. In Sections 4 and 5, we study two specific classes of no-regret learning algorithms and show that the classes of games for which Nash equilibrium convergence can be guaranteed can be expanded if the algorithm is known. In section 6, we use three online marketing examples to validate the proposed algorithms and conditions. We conclude this work in Section 7.

## 2 PROBLEM DEFINITION

Consider an online convex game with $N$ agents, whose goal is to learn their best individual actions that minimize their local loss functions. For each agent $i \in \mathcal{N} = \{1, \ldots, N\}$, denote by $\mathcal{C}_i(x_i, x_{-i}) : \mathcal{X} \to \mathbb{R}$ the individual loss function, where $x_i \in \mathcal{X}_i$ is the action of agent $i$, $x_{-i}$ are the actions of all agents except for agent $i$, and we define $\mathcal{X} = \Pi_{i=1}^N \mathcal{X}_i$ to be the joint action space since each agent takes actions independently. For ease of notation, we collect all agents' actions in a vector $x := (x_1, \ldots, x_N)$. We assume that $\mathcal{C}_i(x)$ is convex in $x_i$ for all $x_{-i} \in \mathcal{X}_{-i}$, where $\mathcal{X}_{-i}$ is the joint action space excluding agent $i$. In addition, we assume that the diameter of the convex set $\mathcal{X}_i$ is bounded by $D$, for all $i = 1, \ldots, N$. The goal of every agent $i$ is to determine the action $x_i$ that minimizes its individual loss function, i.e.,

$$\min_{x_i \in \mathcal{X}_i} \mathcal{C}_i(x_i, x_{-i}). \tag{1}$$

As shown in Rosen (1965), convex games always have at least one Nash equilibrium. In what follows, we denote by $x^*$ a Nash equilibrium of the game (1). Then, for each agent $i$, we have $\mathcal{C}_i(x^*) \leq \mathcal{C}_i(x_i, x_{-i}^*)$, $\forall x_i \in \mathcal{X}_i$, $i \in \mathcal{N}$. At this Nash equilibrium point, agents are strategically stable in the sense that each agent lacks incentives to change its action. Since the agents' loss functions are convex, the Nash equilibrium can also be characterized by the first-order optimality condition, i.e., $\langle \nabla_{x_i} \mathcal{C}_i(x^*), x_i - x_i^* \rangle \geq 0$, $\forall x_i \in \mathcal{X}_i, i \in \mathcal{N}$, where $\nabla_{x_i} \mathcal{C}_i(x)$ is the partial derivative of the loss function with respect to each agent' action. We write $\nabla_i \mathcal{C}_i(x)$ instead of $\nabla_{x_i} \mathcal{C}_i(x)$ whenever it is clear from the context. Throughout the paper, we make the following assumptions on the convex loss functions.

**Assumption 1.** *For each agent $i$, we have that $\mathcal{C}_i(x)$ is $L_0$-Lipschitz continuous in $x$ and $|\mathcal{C}_i(x)| \leq U$.*

**Assumption 2.** *For each agent $i$, we have that $\nabla_i \mathcal{C}_i(x)$ is $L_1$-Lipschitz continuous in $x$ and $\|\nabla_i \mathcal{C}_i(x)\| \leq B$.*

The above assumptions are very common in the literature and hold in many applications, e.g., Cournot Games, retailer pricing games; see Bravo et al. (2018); Duvocelle et al. (2018); Lin et al. (2021).

In general, it is not easy to show convergence to a Nash equilibrium for games with multiple Nash Equilibria. For this reason, recent studies often focus on games that are so-called strongly monotone and are well-known to have a unique Nash equilibrium (Rosen, 1965). In this case, convergence to the Nash equilibrium is shown, e.g., in Drusvyatskiy & Ratliff (2021); Bravo et al. (2018). The game (1) is said to be $m$-strongly monotone if for $\forall x, x' \in \mathcal{X}$ we have that

$$\sum_{i=1}^N \langle \nabla_i \mathcal{C}_i(x) - \nabla_i \mathcal{C}_i(x'), x_i - x_i' \rangle \geq m \|x - x'\|^2. \tag{2}$$

The ability of the agents to efficiently learn their optimal actions can be quantified using the notion of regret that captures the cumulative loss of the learned online actions compared to the best actions in hindsight, and can be formally defined as

$$\text{Reg}_i = \sum_{t=1}^T \mathcal{C}_i(x_t) - \min_{x_i} \sum_{t=1}^T \mathcal{C}_i(x_i, x_{-i,t}), \tag{3}$$

for sequences of actions $\{x_{i,t}\}_{t=1}^T, i = 1, \ldots, N$. An algorithm is said to be no-regret if the regret of each agent is sub-linear in the total number of episodes $T$, i.e., $\text{Reg}_i = \mathcal{O}(T^a), a \in [0, 1), \forall i \in \mathcal{N}$.

In this work, we are interested in understanding for what classes of games and learning algorithms Nash equilibrium convergence can be guaranteed. Specifically, we are interested in understanding whether and for what class of online convex games with continuous action sets no-regret learning converges to a Nash equilibrium regardless of the specific algorithm; see Section 3. Moreover, we are interested in understanding whether and how this class of online convex games can be expanded when the no-regret learning algorithm is known; see Sections 4 and 5.

## 3 NO-REGRET LEARNING CONVERGES TO A NASH EQUILIBRIUM

In this section, we provide our main result which shows that any no-regret learning algorithm can guarantee Nash equilibrium convergence for the class of $m$-strongly monotone games that satisfy an additional condition on the parameter $m$. We start with single-agent learning and then extend it to multi-agent games.

### 3.1 SINGLE-AGENT LEARNING

Consider a single agent whose goal is to minimize its convex loss function $\mathcal{C}(x)$ by optimizing its action $x \in \mathcal{X}$. Let $x^* = \operatorname{argmin}_x \mathcal{C}(x)$. Suppose that an online algorithm generates a sequence $\{x_t\}_{t=1}^T$, and the regret is defined as $\text{Reg} = \sum_{t=1}^T \mathcal{C}(x_t) - \mathcal{C}(x^*)$. In the following lemma, we show that strong convexity of the loss function $\mathcal{C}(x)$ guarantees that no-regret learning leads to Nash equilibrium convergence.

**Lemma 1.** *Suppose that the loss function $\mathcal{C}(x)$ is $m$-strongly convex in $x$ with the parameter $m > 0$. If the regret is sub-linear in $T$ such that $\text{Reg} = \mathcal{O}(T^a)$ with $a \in [0, 1)$, then we have $\sum_{t=1}^T \|x_t - x^*\|^2 = \mathcal{O}(T^a)$.*

*Proof.* The strong convexity of the loss function $\mathcal{C}(x)$ implies that $\mathcal{C}(x) - \mathcal{C}(x^*) \geq \frac{m}{2} \|x - x^*\|^2, \forall x \in \mathcal{X}$. Substituting in this inequality the agent's action $x_t$ at time $t$ and summing over $t$, we obtain that $\text{Reg} = \sum_{t=1}^T (\mathcal{C}(x_t) - \mathcal{C}(x^*)) \geq \frac{m}{2} \sum_{t=1}^T \|x_t - x^*\|^2$. The result follows by the fact that $\frac{m}{2}$ is a constant that does not depend on $T$. $\square$

Lemma 1 shows that no-regret learning for single-agent learning guarantees time-averaged convergence to the stable point when the loss function is strongly convex, i.e., $\frac{1}{T} \sum_{t=1}^T \|x_t - x^*\|^2 \to 0$ as $T \to \infty$. It's well known that strong convexity is equivalent to the condition

$$\langle \nabla \mathcal{C}(x) - \nabla \mathcal{C}(x'), x - x' \rangle \geq m \|x - x'\|^2. \tag{4}$$

Moreover, note that the strong monotonicity condition (2) is equivalent to (4) in the case of single-agent learning, i.e., when $N = 1$. This observation inspires the following analysis for multi-agent games.

### 3.2 MULTI-AGENT GAMES

As discussed in Section 3.1, strong monotonicity is sufficient for the convergence of no-regret single-agent learning to an equilibrium point. However, the extension of this result from single-agent learning to multi-agent games is non-trivial, due to the structure of the agents' loss functions that are coupled by the other agents' actions. Besides, every time an agent updates its action, the other agents also react to this change. Therefore, since the actions $x_{-i,t}$ also change, the function $\mathcal{C}_i(\cdot, x_{-i,t})$ becomes non-stationary from the perspective of agent $i$. In the following result, we show that if the game is sufficiently strongly monotone, no-regret learning can still guarantee Nash equilibrium convergence.

**Theorem 1.** *Suppose that the game (1) is $m$-strongly monotone and $\nabla_i \mathcal{C}_i(x_i, x_{-i})$ is $L$-Lipschitz continuous in $x_{-i}$ for every $x_i \in \mathcal{X}_i$. Suppose that an algorithm generates the action sequences $\{x_{i,t}\}$, $i = 1, \ldots, N$, and that the regret of each agent satisfies $\text{Reg}_i = \sum_{t=1}^T \left( \mathcal{C}_i(x_t) - \mathcal{C}_i(y_i^*, x_{-i,t}) \right) = \mathcal{O}(T^a)$, $\forall i = 1, \ldots, N$, where $y_i^* = \operatorname{argmin}_{y_i} \sum_{t=1}^T \mathcal{C}_i(y_i, x_{-i,t})$ and $a \in [0, 1)$. Let $x^*$ denote the unique Nash equilibrium and $y^* = (y_1^*, \ldots, y_N^*)$. Then, the following hold:*

1. $\|x^* - y^*\| \leq \frac{NL}{mT} \sum_{t=1}^{T} \|x_t - x^*\|$;

2. If $m - 2L\sqrt{N-1} > 0$, then $\sum_{t=1}^{T} \|x_t - x^*\|^2 = \mathcal{O}(T^a)$.

The proof can be found in Appendix 8.2. Theorem 1 implies that any no-regret algorithm can lead to Nash equilibrium whenever $m - 2L\sqrt{N-1} > 0$. Note that the condition $m - 2L\sqrt{N-1} > 0$ always holds for single-agent learning where $N = 1$, as long as $m > 0$, which coincides with Lemma 1.

**Remark 1.** *Recall that $L_1$ and $L$ are the Lipschitz constants of the function $\nabla_i \mathcal{C}_i(x)$ with respect to $x$ and $x_{-i}$, respectively. From the definitions we can conclude that $L \leq L_1$. The Lipschitz constant $L_1$ provides an upper bound on the variation of gradients and is always greater than the strongly monotone parameter $m$, which provides a lower bound, i.e., $m \leq L_1$. However, it is still likely to have $m - 2L\sqrt{N-1} > 0$. For example, in the extreme case where $\mathcal{C}_i$ only depends on $x_i$, we have that $L = 0$ and thus the condition naturally holds as long as $m > 0$.*

**Remark 2.** *We provide here some intuition regarding regarding the condition $m - 2L\sqrt{N-1} > 0$. Rearranging the terms in the inequality gives $L < \frac{m}{2\sqrt{N-1}}$. Recall that $L$ is the Lipschitz constant of the function $\nabla_i \mathcal{C}_i(x_i, x_{-i})$ with respect to $x_{-i}$, which can be interpreted as a bound on the maximum influence of the other agents' actions. We need this influence of the other agents' actions and, therefore, $L$, to be small for the game to converge. On the other hand, the presence of multiple agents ($N$ is large) makes the game increasingly involved, which restricts the class of games for which no-regret learning converges to a Nash Equilibruim.*

In general, it is easier to analyze the regret of an algorithm compared to analyzing the Nash equilibrium convergence. Theorem 1 provides an alternative way to analyze the Nash equilibrium convergence: As long as we can show an algorithm is no-regret and a game is $m$-strongly monotone with $m - 2L\sqrt{N-1} > 0$, we can directly obtain Nash equilibrium convergence.

# 4 SUFFICIENT CONDITIONS FOR CONVERGENCE OF GRADIENT DESCENT ALGORITHMS

In this section, we provide sufficient conditions for the convergence of multi-agent games to a Nash equilibrium with the additional knowledge that the no-regret learning algorithm is a gradient-descent (GD) algorithm. Specifically, we investigate two types of GD algorithms: a first-order algorithm and a zeroth-order algorithm. Note that, under Assumptions 1 and 2, it can be easily verified that both first-order and zeroth-order GD algorithms ensure that the regret of each agent in (3) is sub-linear in $T$; see Hazan et al. (2016); Flaxman et al. (2004). In what follows, we show that Nash equilibrium convergence can be guaranteed for both algorithms as long as $m > 0$.

## 4.1 FIRST-ORDER ALGORITHMS

We first consider the case where the agents have access to their gradient information. Specifically, we assume that each agent can obtain an unbiased gradient estimate $G_i$ of $\nabla_i \mathcal{C}_i(x)$ with finite variance given the joint action profile $x$, where $\mathbb{E}[G_i] = \nabla_i \mathcal{C}_i(x)$ and $\mathbb{E}[G_i - \nabla_i \mathcal{C}_i(x)]^2 \leq \sigma^2$. During learning, we assume that each agent performs the following action update

$$x_{i,t+1} = \mathcal{P}_{\mathcal{X}_i}(x_{i,t} - \eta_t G_{i,t}), \tag{5}$$

where $\mathbb{E}[G_{i,t}] = \nabla_i \mathcal{C}_i(x_t)$ and $\mathcal{P}_{\mathcal{X}_i}$ projects the agent's actions to its action space $\mathcal{X}_i$. In the following result, we present the Nash equilibrium convergence analysis of the first-order GD algorithm (5).

**Theorem 2.** *Let Assumptions 1 and 2 hold. Suppose that the game (1) is $m$-strongly monotone with parameter $m > 0$. Then, the first-order GD algorithm (5) with $\eta_t = \frac{1}{mt}$ satisfies that*

$$\mathbb{E}\|x_T - x^*\|^2 = \mathcal{O}(T^{-1}). \tag{6}$$

See Appendix 8.3 for the detailed proof. Note that equation (6) implies that there exists a constant $C_0 > 0$ such that $\mathbb{E}\|x_t - x^*\|^2 \leq C_0 t^{-1}$ for $\forall t = 1, \ldots, T$. Summing over $t$

we get $\mathbb{E} \sum_{t=1}^{T} \|x_t - x^*\|^2 \leq \sum_{t=1}^{T} C_0 t^{-1} \leq C_0 \left(1 + \int_1^T \frac{1}{t} dt\right) \leq C_0(1 + \ln T)$, and thus $\mathbb{E} \sum_{t=1}^{T} \|x_t - x^*\|^2 = \mathcal{O}(\ln T)$. Therefore, the last iteration convergence implies the time-averaged convergence of the algorithm.

## 4.2 ZEROTH-ORDER ALGORITHMS

In many games, the agents have access to limited information and cannot observe the other agents' actions. For example, in the Cournot competition and Kelly auction games (Bravo et al., 2018; Lin et al., 2021), each company (agent) is not willing to share its strategy (action) with the rivals and would rather keep it as a secret. In this case, first-order gradient information is not available since it depends on the joint action of all agents. Instead, here, we assume that the agents can only access their own loss function evaluation, which is also referred to as bandit feedback. In this case, a common and effective approach to estimate the unknown gradient is to utilize zeroth-order methods. Specifically, at each episode, the agents perturb their actions $x_{i,t}$ by an amount $\delta u_{i,t}$, where $u_{i,t} \in \mathbb{S}^{d_i}$ is a random variable sampled from the unit sphere $\mathbb{S}^{d_i} \subset \mathbb{R}^{d_i}$ and $\delta$ is the size of this perturbation. Then, the agents play their perturbed actions $\hat{x}_{i,t} = x_{i,t} + \delta u_{i,t}$ and receive as feedback their local losses $\mathcal{C}_i(\hat{x}_t)$. Using this information, every agent constructs its own gradient estimate $g_{i,t} = \frac{d_i}{\delta} \mathcal{C}_i(\hat{x}_t) u_{i,t}$, and performs the following update

$$x_{i,t+1} = \mathcal{P}_{(1-\delta)\mathcal{X}_i}(x_{i,t} - \eta_t g_{i,t}), \tag{7}$$

where the projection set $(1 - \delta)\mathcal{X}_i$ is to ensure the feasibility of the next played action.

To facilitate the analysis, we define the $\delta$-smoothed function $\mathcal{C}_i^\delta(x) = \mathbb{E}_{w_i \sim \mathbb{B}_i, u_{-i} \sim \mathbb{S}_{-i}}[\mathcal{C}_i(x_i + \delta w_i, x_{-i} + \delta u_{-i})]$, where $\mathbb{S}_{-i} = \Pi_{j \neq i} \mathbb{S}_j$, and $\mathbb{B}_i$, $\mathbb{S}_i$ denote the unit ball and unit sphere in $\mathbb{R}^{d_i}$, respectively. As shown in Drusvyatskiy & Ratliff (2021); Bravo et al. (2018), the function $\mathcal{C}_i^\delta(x)$ satisfies the following properties.

**Lemma 2.** *Let Assumptions 1 and 2 hold. Then we have that*

1. $\mathcal{C}_i^\delta(x_i, x_{-i})$ *is convex in* $x_i$;

2. $\mathcal{C}_i^\delta(x)$ *is* $L_0$-*Lipschitz continuous in* $x$;

3. $|\mathcal{C}_i^\delta(x) - \mathcal{C}_i(x)| \leq \delta L_0 \sqrt{N}$;

4. $\mathbb{E}[\frac{d_i}{\delta} \mathcal{C}_i(\hat{x}_t) u_{i,t}] = \nabla_i \mathcal{C}_i^\delta(x_t)$.

Note that, since the game with the loss function $\mathcal{C}_i(x)$ is assumed strongly monotone, it has a unique Nash equilibrium. However, without assuming that the Jacobian of $\nabla_i \mathcal{C}_i(x)$ is Lipschitz continuous as in Drusvyatskiy & Ratliff (2021), the smoothed game defined by the loss function $\mathcal{C}_i^\delta(x)$ over the set $(1-\delta)\mathcal{X}$ is possibly not strongly monotone and can have multiple Nash Equlibria. As we discuss below, even if the smoothed game has multiple Nash Equilibria, it is still possible to show that the original game converges to a Nash equilibrium. To do so, we first provide a lemma that bounds the distance between the Nash Equilibria of the smoothed game. In this lemma, for a given $\delta$, we denote by $\mathcal{A}^\delta$ the set of Nash Equilibria of the smoothed game with loss $\mathcal{C}_i^\delta(x)$ contained in the set $(1 - \delta)\mathcal{X}$. Since $\mathcal{C}_i^\delta(x)$ is convex in $x_i$, we know that there exists at least one Nash equilibrium in the smoothed game, i.e., $\mathcal{A}^\delta \neq \emptyset$.

**Lemma 3.** *Suppose that Assumptions 1 and 2 hold and that the game* (1) *is* $m$-*strongly monotone. Moreover, assume that the smoothed game with losses* $\mathcal{C}_i^\delta(x)$ *has multiple Nash Equilibria in the set* $(1 - \delta)\mathcal{X}$. *Then the distance between any arbitrary two Nash Equilibria is bounded by* $\frac{2L_1 \delta N}{m}$.

The detailed proof can be found in Appendix 8.4.

Lemma 3 states that, although there may exist multiple Nash Equilibria in the smoothed game, the distance between these Nash Equilibria can be bounded by the parameter $\delta$. Based on this observation, we can further bound the distance between the Nash equilibrium $x^*$ of the original game and the Nash Equilibria of the smoothed game, as shown in the following lemma.

**Lemma 4.** *Suppose that Assumptions 1 and 2 hold and that the game* (1) *is $m$-strongly monotone. Then, any Nash equilibrium $x^{\delta,j} \in \mathcal{A}^\delta$ of the smoothed game satisfies that*

$$\left\| x^* - x^{\delta,j} \right\| \leq \delta \left( \left( 1 + \frac{L_1 \sqrt{N}}{m} \right) \|x^*\| + \frac{L_1 N}{m} \right). \tag{8}$$

We provide the detailed proof in Appendix 8.5.

Lemma 4 shows that even if the smoothed game is not strongly monotone and possibly has multiple Nash Equilibria, we can still upper bound the distance between these Nash Equilibria and the Nash equilibrium of the original game. The result in Lemma 4 is the same as that in Drusvyatskiy & Ratliff (2021), with the difference that Drusvyatskiy & Ratliff (2021) make the additional assumption that the Jacobian of the gradient of the cost function is Lipschitz continuous. Here, we show that this assumption is not needed and the statement of the lemma still holds. We now present the main result.

**Theorem 3.** *Let Assumptions 1 and 2 hold. Suppose that the game* (1) *is $m$-strongly monotone with parameter $m > 0$. Then, the zeroth-order GD algorithm* (7) *with $\eta_t = \frac{1}{mt}$, and $\delta = T^{-\frac{1}{3}}$ satisfies that*

$$\mathbb{E} \left\| x_T - x^* \right\|^2 = \mathcal{O}(T^{-\frac{1}{3}}). \tag{9}$$

See Appendix 8.6 for the proof.

Since $\sum_{t=1}^{T} t^{-\frac{1}{3}} \leq 1 + \int_1^T \frac{1}{t^{\frac{1}{3}}} dt \leq 1 + \frac{3}{2} t^{\frac{2}{3}} \big|_1^T \leq 1 + \frac{3}{2} T^{\frac{2}{3}}$, we obtain that $\mathbb{E} \sum_{t=1}^{T} \| x_t - x^* \|^2 = \mathcal{O}(T^{\frac{2}{3}})$. Thus the time-averaged convergence of the game is guaranteed.

## 5 SUFFICIENT CONDITIONS FOR CONVERGENCE OF THE BEST RESPONSE ALGORITHM

In the previous section, we showed that gradient descent algorithms guarantee convergence of $m$-strongly monotone games to a Nash equilibrium provided that $m > 0$. In this section, we provide sufficient conditions for Nash equilibrium convergence for the best response algorithm. The best response is a common strategy in the game theory literature, especially that on fully competitive games, that produces the most favorable outcome given the other agents plays. In continuous games (1), the best response is defined as:

$$x_{i,t+1} = \underset{x_i \in \mathcal{X}_i}{\arg\min} \, \mathcal{C}_i(x_i, x_{-i,t}), \tag{10}$$

i.e., each agent takes the action that provides the best response to the other agents' actions from the previous episode. The convergence analysis of the best response algorithm is presented below.

**Theorem 4.** *Suppose that the game* (1) *is $m$-strongly monotone with parameter $m > L\sqrt{N-1}$. Then the best response algorithm* (10) *satisfies that*

$$\left\| x_T - x^* \right\|^2 \leq \rho^T \left\| x_0 - x^* \right\|^2, \tag{11}$$

*where $\rho := \frac{L^2(N-1)}{m^2}$.*

The proof is provided in Appendix 8.7. From the inequality (11), it is easy to obtain that $\sum_{t=1}^{T} \| x_t - x^* \|^2 \leq \sum_{t=1}^{T} \rho^t \| x_0 - x^* \|^2 \leq \frac{\|x_0 - x^*\|^2}{1-\rho} = \mathcal{O}(1)$. Theorem 4 shows that the best response algorithm converges to the Nash equilibrium at a linear rate. Indeed, it is a no-regret learning algorithm for each agent as well, as shown in Theorem 5 in Appendix 8.8.

To the best of our knowledge, this is the first effort to provide a sufficient condition ($m > L\sqrt{N-1}$) under which the best-response algorithm achieves Nash equilibrium convergence in convex games. Moreover, we experimentally show that when $m > L\sqrt{N-1}$ does not hold, the best-response algorithm may lead to cycles, which further supports our theoretical results.

## 6 NUMERICAL EXPERIMENTS

In this section, we illustrate and compare the proposed algorithms on the Cournot game problems. Additional experiments on retailer pricing games and Kelly auction games can be found in Appendix 8.9 and 8.10, respectively. We show that, by defining sufficient conditions for Nash equilibrium convergence that depend only the properties of the game, i.e., the parameter $m$, and not the learning algorithm used to solve it, our analysis allows to identify classes of games for which no-regret learning guarantees convergence to a Nash equilibrium without analyzing specific algorithms or identify specific no-regret learning algorithms with no guaranteed convergence to a Nash equilibrium.

We first consider a Cournot game with two agents whose goal is to minimize their local loss by appropriately setting the production quantity $x_i$, $i = 1, 2$. The loss function of each agent is given by $\mathcal{C}_i(x) = x_i(\frac{a_i x_i}{2} + b_i x_{-i} - e_i) + 1$, where $a_i > 0$, $b_i$, $e_i$ are constant parameters, and $x_{-i}$ denotes the production quantity of the opponent of agent $i$. It is easy to show that $\nabla_i \mathcal{C}_i(x) = a_i x_i + b_i x_{-i} - e_i$. Recalling that $L$ is the Lipschitz constant of the function $\nabla_i \mathcal{C}_i(x)$ with respect to $x_{-i}$, we have $L = \max\{b_1, b_2\}$. Define $g(x) = (\nabla_1 \mathcal{C}_1(x), \nabla_2 \mathcal{C}_2(x))$ and let $G(x)$ denote the Jacobian of $g(x)$, i.e., $G(x) = [a_1, b_1; b_2, a_2]$. According to Rosen (1965), the strong monotonicity parameter $m$ in this simple example coincides with the smallest eigenvalue of the matrix $\frac{G(x)+G'(x)}{2}$. In what follows, we verify and compare the effectiveness of the first-order gradient descent algorithm (FO), the zeroth-order gradient descent algorithm (ZO), and the best response algorithm (BR), analyzed in Sections 4 and 5. To apply the first-order algorithm, we assume that the gradient is subject to a noise sampled from the normal distribution. For all algorithms, the feasible set is defined as $\mathcal{X}_i = \{x_i | x_i \in [0, 3]\}$.

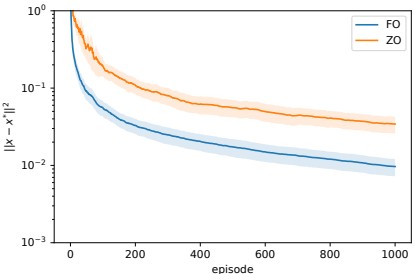 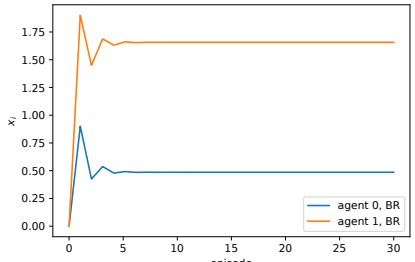

(a) Error to Nash equilibrium of the first-order method (FO) and the zeroth-order method (ZO).

(b) Action values of the best response algorithm (BR).

Figure 1: Cournot game when $m > L\sqrt{N-1}$.

We validate our methods for two different selections of parameters. First, we select the parameters $(a_1, a_2) = (2, 1)$, $(b_1, b_2) = (0.5, 0.5)$, $(e_1, e_2) = (1.8, 1.9)$. Using these parameters, we get that $m = 0.79$, $L = 0.5$, and, therefore, $m > L\sqrt{N-1}$, so that the sufficient conditions for all three algorithms are satisfied. The results are shown in Figure 1, where the solid lines and shades are averages and $\pm$ standard deviations over 60 runs, respectively. Specifically, in Figure 1 (a), the two gradient-descent algorithms both converge to the Nash equilibrium, and the first-order algorithm outperforms the zeroth-order algorithm in terms of convergence speed and variance. Figure 1 (b) illustrates the action updates of the best response algorithm. Since the best response algorithm performs the action update in a way that each agent selects the optimal actions against other agents' actions, this algorithm converges very fast to the Nash equilibrium point $(0.4857, 1.657)$.

Next we select the parameters $(a_1, a_2) = (2, 1)$, $(b_1, b_2) = (-1.5, 1.5)$, $(e_1, e_2) = (1.8, 1.9)$. In this case, we have that $m = 1$, $L = 1.5$, and, therefore, $m < L\sqrt{N-1}$. As a result, the sufficient condition for the best response algorithm is not satisfied. The simulation results in this case are presented in Figure 2. When $m < L\sqrt{N-1}$, the first-order and zeroth-order gradient descent methods still converge to the Nash equilibrium, as shown in Figure 2 (a). Figure 2 (b) shows that the agents' actions of the best response algorithm oscillate in a regular pattern after some episodes and fail to converge to the Nash equilibrium. Therefore, the best response algorithm does not guarantee convergence to the Nash equilibrium when the sufficient condition is not satisfied.

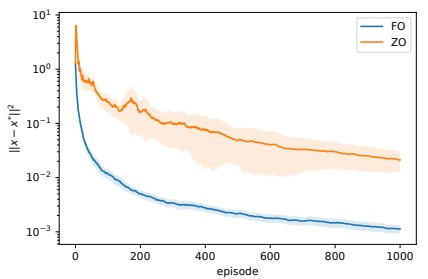 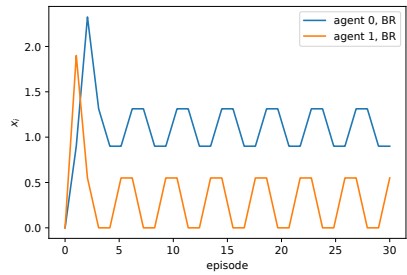

(a) Error to Nash equilibrium of the first-order method (FO) and the zeroth-order method (ZO).

(b) Action values of the best response algorithm (BR).

Figure 2: Cournot game when $0 < m < L\sqrt{N-1}$.

We also consider a Cournot game with $5$ agents, $i = 1, 2, 3, 4, 5$. The loss function of each agent is $C_i(x) = x_i(\frac{a_i x_i}{2} + b_i \sum_{j \neq i} x_j - e_i) + 1$, where $a = [2, 2, 1.5, 1.8, 2]$, $b = [0.2, 0.3, 0.3, 0.2, 0.3]$, $e = [1.8, 1.9, 1.5, 1.6, 1.8]$. In this case $m = 1.2844$, $L = 0.6$, and therefore $m > L\sqrt{N-1}$ so that the sufficient conditions are satisfied for all three algorithms. The Nash equilibrium is $x^* = [0.672, 0.597, 0.512, 0.631, 0.538]$. Figure 3 shows that all three algorithms converge to this Nash equilibrium.

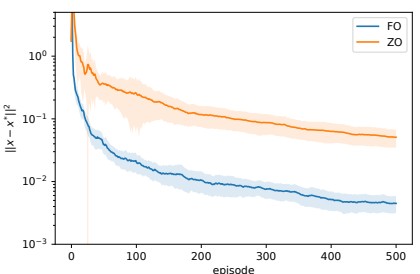 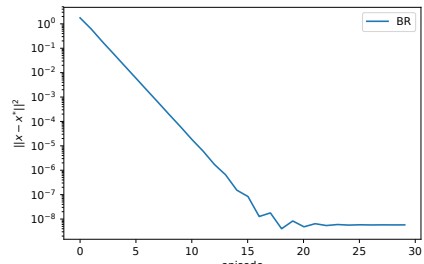

(a) Error to Nash equilibrium of the first-order method (FO) and the zeroth-order method (ZO).

(b) Error to Nash equilibrium of the best response algorithm (BR).

Figure 3: Cournot game with multiple agents $N = 5$ when $m > L\sqrt{N-1}$.

## 7 CONCLUSION

In this work, we study the connection between no-regret learning and time-average Nash equilibrium convergence for the class of strongly monotone games. Specifically, we provided a sufficient condition on the class of strongly monotone games for which any no-regret learning algorithm leads to Nash equilibrium convergence. Moreover, we showed that the class of these games can be expanded when additional information about a specific no-regret algorithm is considered, including the first-order and zeroth-order gradient descent algorithms and the best response algorithm. We numerically validated our theoretical results for different games that belong to different classes, including Cournot games, retailer pricing games, and Kelly auction games. Compared to existing literature that analyzes the regret and Nash equilibrium convergence for specific algorithms and for assumptions that depend on the specific choice of algorithms and games, here we proposed a different approach that focuses on understanding the fundamental relationship between no-regret learning and Nash equilibrium convergence, regardless of the specific learning algorithm and based only on the game type.

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

## 8 APPENDIX

### 8.1 AUXILIARY LEMMAS

**Lemma 5.** *(Bravo et al. (2018)) Let $a_n$, $n = 1, 2, \ldots$, be a non-negative sequence such that*

$$a_{n+1} \leq a_n(1 - \frac{A}{n}) + \frac{E}{n^{1+q}}, \tag{12}$$

*where $q > 0$, and $A, E > 0$. Then, assuming $A > q$, we have that*

$$a_n \leq \frac{E}{(A-q)n^q} + \mathcal{O}\left(\frac{1}{n^q}\right). \tag{13}$$

### 8.2 PROOF OF THEOREM 1:

We first show a useful lemma that lays the foundation for the subsequent analysis.

**Lemma 6.** *If the game with the cost $\mathcal{C}_i(x_i, x_{-i})$ is strongly monotone with parameter $m$, then the function $\mathcal{C}_i(z, x_{-i})$ is strongly convex in $z$.*

*Proof.* From the strong monotonicity condition (2), it follows that

$$\langle \nabla_i \mathcal{C}_i(z, x_{-i}) - \nabla_i \mathcal{C}_i(z', x_{-i}), z - z' \rangle \geq m \|z - z'\|^2. \tag{14}$$

Define $h_i(z) := \mathcal{C}_i(z, x_{-i})$. Then, inequality (14) becomes $\langle \nabla h_i(z) - \nabla h_i(z'), z - z' \rangle \geq m \|z - z'\|^2$. The result follows from the fact that a differentiable function $f$ is strongly convex if and only if its domain is convex and $\langle \nabla f(x) - \nabla f(x'), x - x' \rangle \geq m \|x - x'\|^2$. $\square$

Using Lemma 6, we have that the function $\mathcal{C}_i(z, x_{-i})$ is strongly convex in $z$, which means that $\mathcal{C}_i(y_i', x_{-i}) \geq \mathcal{C}_i(y_i, x_{-i}) + \langle \nabla_i \mathcal{C}_i(y_i, x_{-i}), y_i' - y_i \rangle + \frac{m}{2} \|y_i' - y_i\|^2$ for any fixed $x_{-i} \in \mathcal{X}_{-i}$. Recall that $y_i^* = \mathrm{argmin}_{y_i} \sum_{t=1}^{T} \mathcal{C}_i(y_i, x_{-i,t})$ and $x^*$ denotes the Nash equilibrium of the game (1). Since the sum operator preserves convexity, we have that $\sum_t \mathcal{C}_i(y_i, x_{-i,t})$ is also convex in $y_i$. Moreover, from the necessary condition of optimality, we have that $\sum_t \langle \nabla_i \mathcal{C}_i(y_i^*, x_{-i,t}), y_i - y_i^* \rangle \geq 0$ for $\forall y_i \in \mathcal{X}_i$. Since $x_i^* \in \mathcal{X}_i$, replacing $y_i$ with $x_i^*$ gives that

$$\sum_t \langle \nabla_i \mathcal{C}_i(y_i^*, x_{-i,t}), x_i^* - y_i^* \rangle \geq 0. \tag{15}$$

Next, we analyze the distance between $x^*$ and $y^*$, where $y^* = (y_1^*, \ldots, y_1^*)$. By strong monotonicity of $\mathcal{C}_i(x_i, x_{-i})$, we have that $\langle \nabla_i \mathcal{C}_i(y_i^*, x_{-i,t}) - \nabla_i \mathcal{C}_i(x_i^*, x_{-i,t}), y_i^* - x_i^* \rangle \geq m \|y_i^* - x_i^*\|^2$. Summing this inequality over $t = 1 \ldots, T$, and combining the inequality (15), we have that

$$mT \|x_i^* - y_i^*\|^2 \leq \sum_t \langle \nabla_i \mathcal{C}_i(y_i^*, x_{-i,t}) - \nabla_i \mathcal{C}_i(x_i^*, x_{-i,t}), y_i^* - x_i^* \rangle \leq \sum_t \langle -\nabla_i \mathcal{C}_i(x_i^*, x_{-i,t}), y_i^* - x_i^* \rangle. \tag{16}$$

Summing the inequality (16) over $i = 1, \ldots, N$, we further obtain

$$mT \|x^* - y^*\|^2 \leq \sum_i \sum_t \langle -\nabla_i \mathcal{C}_i(x_i^*, x_{-i,t}), y_i^* - x_i^* \rangle$$

$$= \sum_i \sum_t \langle \nabla_i \mathcal{C}_i(x^*) - \nabla_i \mathcal{C}_i(x_i^*, x_{-i,t}), y_i^* - x_i^* \rangle - \sum_i \sum_t \langle \nabla_i \mathcal{C}_i(x^*), y_i^* - x_i^* \rangle$$

$$\leq \sum_i \sum_t \langle \nabla_i \mathcal{C}_i(x^*) - \nabla_i \mathcal{C}_i(x_i^*, x_{-i,t}), y_i^* - x_i^* \rangle$$

$$\leq \sum_i \sum_t L \|x_{-i,t} - x_{-i}^*\| \|x_i^* - y_i^*\|$$

$$\leq \sum_i \sum_t L \|x_t - x^*\| \|x^* - y^*\|$$

$$\leq NL \|x^* - y^*\| \sum_t \|x_t - x^*\|,$$

where the second inequality is due to the fact that $x^*$ is a Nash equilibrium and the third inequality follows from the Lipschitz continuous property of the function $\nabla_i \mathcal{C}_i$ with respect to $x_{-i}$. Rearranging the terms in the above inequality, we obtain the first of the two theorem statements.

Next, we analyze the regret of the action sequence $\{x_{i,t}\}$ for $i = 1, ..., N$. By the definition of regret in (3), we have that

$$
\begin{aligned}
\text{Regret}_i &= \sum_t \mathcal{C}_i(x_t) - \sum_t \mathcal{C}_i(y_i^*, x_{-i,t}) \\
&= \sum_t \Big( \mathcal{C}_i(x_t) - \mathcal{C}_i(x_i^*, x_{-i,t}) \Big) + \sum_t \Big( \mathcal{C}_i(x_i^*, x_{-i,t}) - \mathcal{C}_i(y_i^*, x_{-i,t}) \Big) \\
&\geq \sum_t \Big( \langle \nabla_i \mathcal{C}_i(x_i^*, x_{-i,t}), x_{i,t} - x_i^* \rangle + \frac{m}{2} \|x_i^* - x_{i,t}\|^2 \Big) + \sum_t \Big( \langle \nabla_i \mathcal{C}_i(y_i^*, x_{-i,t}), x_i^* - y_i^* \rangle + \frac{m}{2} \|y_i^* - x_i^*\|^2 \Big) \\
&\geq \sum_t \Big( \langle \nabla_i \mathcal{C}_i(x_i^*, x_{-i,t}), x_{i,t} - x_i^* \rangle + \frac{m}{2} \|x_i^* - x_{i,t}\|^2 \Big) + \frac{mT}{2} \|y_i^* - x_i^*\|^2, \quad (17)
\end{aligned}
$$

where the first inequality follows from the strong convexity of the function $\mathcal{C}_i(z, x_{-i})$ with respect to $z$ for any $x_{-i} \in \mathcal{X}_{-i}$ and the second inequality follows from the necessary condition of optimality. Summing the regret in (17) over $i = 1, \ldots, N$, we have that

$$
\begin{aligned}
\sum_i \text{Regret}_i &= \sum_i \sum_t \Big( \mathcal{C}_i(x_t) - \mathcal{C}_i(y_i^*, x_{-i,t}) \Big) \\
&\geq \sum_i \sum_t \langle \nabla_i \mathcal{C}_i(x_i^*, x_{-i,t}), x_{i,t} - x_i^* \rangle + \frac{m}{2} \sum_t \|x^* - x_t\|^2 + \frac{mT}{2} \|y^* - x^*\|^2 \\
&= \sum_i \sum_t \langle \nabla_i \mathcal{C}_i(x_i^*, x_{-i,t}) - \nabla_i \mathcal{C}_i(x^*), x_{i,t} - x_i^* \rangle + \sum_t \sum_i \langle \nabla_i \mathcal{C}_i(x^*), x_{i,t} - x_i^* \rangle + \frac{mT}{2} \|y^* - x^*\|^2 \\
&\quad + \frac{m}{2} \sum_t \|x^* - x_t\|^2 \\
&\geq \sum_i \sum_t \langle \nabla_i \mathcal{C}_i(x_i^*, x_{-i,t}) - \nabla_i \mathcal{C}_i(x^*), x_{i,t} - x_i^* \rangle + \frac{m}{2} \sum_t \|x^* - x_t\|^2 + \frac{mT}{2} \|y^* - x^*\|^2 \\
&\geq \sum_i \sum_t (-L) \|x_{-i,t} - x_{-i}^*\| \|x_{i,t} - x_i^*\| + \frac{m}{2} \sum_t \|x^* - x_t\|^2 + \frac{mT}{2} \|y^* - x^*\|^2, \quad (18)
\end{aligned}
$$

where the second to the last inequality follows from the fact that $x^*$ is a Nash equilibrium and, therefore, satisfies $\sum_i \langle \nabla_i \mathcal{C}_i(x^*), x_i - x_i^* \rangle \geq 0$ for $\forall x \in \mathcal{X}$, and the last inequality is due to the fact that $\nabla_i \mathcal{C}_i(x)$ is Lipschitz continuous in $x_{-i}$. Applying the inequality $ab \leq \frac{1}{2\lambda} a^2 + \frac{\lambda}{2} b^2, \forall \lambda > 0$ to the term $\|x_{-i,t} - x_{-i}^*\| \|x_{i,t} - x_i^*\|$ in (18), we have that

$$
\begin{aligned}
\sum_i \text{Regret}_i &\geq -L \sum_i \sum_t \Big( \frac{1}{2\lambda} \|x_i^* - x_{i,t}\|^2 + \frac{\lambda}{2} \|x_{-i}^* - x_{-i,t}\|^2 \Big) + \frac{m}{2} \sum_t \|x^* - x_t\|^2 + \frac{mT}{2} \|y^* - x^*\|^2 \\
&\geq -L \sum_t \Big( \frac{1}{2\lambda} \|x^* - x_t\|^2 + \frac{\lambda}{2}(N-1) \|x^* - x_t\|^2 \Big) + \frac{m}{2} \sum_t \|x^* - x_t\|^2 + \frac{mT}{2} \|y^* - x^*\|^2, \quad (19)
\end{aligned}
$$

where the last inequality follows from the fact that $\sum_i \|x_{-i}^* - x_{-i,t}\|^2 = (N-1) \|x^* - x_t\|^2$. Inequality (19) holds for $\forall \lambda > 0$. Substituting $\lambda = (N-1)^{-\frac{1}{2}}$ in (19), we get that

$$
\begin{aligned}
\sum_i \text{Regret}_i &\geq -L\sqrt{N-1} \sum_t \|x^* - x_t\|^2 + \frac{m}{2} \sum_t \|x^* - x_t\|^2 + \frac{mT}{2} \|y^* - x^*\|^2 \\
&\geq \frac{m - 2L\sqrt{N-1}}{2} \sum_t \|x_t - x^*\|^2 + \frac{mT}{2} \|y^* - x^*\|^2. \quad (20)
\end{aligned}
$$

Since we assume that the learning algorithm is no-regret, we have that $\frac{m-2L\sqrt{N-1}}{2} \sum_t \|x_t - x^*\|^2 + \frac{m}{2} \sum_t \|x^* - x_t\|^2 \leq \sum_i \text{Regret}_i = O(T^a)$. Given that the two terms in the left hand of this

inequality are always positive, we conclude that $\sum_t \|x^* - x_t\|^2 = O(T^a)$ and $\frac{mT}{2} \|y^* - x^*\|^2 = O(T^a)$. The proof is complete.

### 8.3 PROOF OF THEOREM 2:

According to Theorem 2 in Rosen (1965), strongly monotone games have a unique Nash equilibrium. Let $x^* = (x_i^*, x_{-i}^*)$ denote the Nash equilibrium of the game (1). Using the update equation (5), we have that

$$
\begin{aligned}
\|x_{i,t+1} - x_i^*\|^2 &= \|\mathcal{P}_{\mathcal{X}_i}(x_{i,t} - \eta_t G_{i,t}) - x_i^*\|^2 \\
&\leq \|x_{i,t} - x_i^* - \eta_t G_{i,t}\|^2 \\
&\leq \|x_{i,t} - x_i^*\|^2 + \eta_t^2 \|G_{i,t}\|^2 - 2\eta_t \langle G_{i,t}, x_{i,t} - x_i^* \rangle,
\end{aligned}
$$

where the first inequality follows from the facts that $\mathcal{P}_{\mathcal{X}_i}(x_i^*) = x_i^*$ and $\|\mathcal{P}_{\mathcal{X}_i}(z)\| \leq \|z\|$ for any vector $z$. Taking the expectation of both sides of the above inequality, we get

$$
\begin{aligned}
\mathbb{E}\|x_{i,t+1} - x_i^*\|^2 &\leq \mathbb{E}\Big[ \|x_{i,t} - x_i^*\|^2 + \eta_t^2 \|G_{i,t}\|^2 - 2\eta_t \langle G_{i,t}, x_{i,t} - x_i^* \rangle \Big] \\
&\leq \mathbb{E}\|x_{i,t} - x_i^*\|^2 + \eta_t^2 \|\nabla_i \mathcal{C}_i(x_t)\|^2 + \eta_t^2 \sigma^2 - 2\eta_t \langle \nabla_i \mathcal{C}_i(x_t), x_{i,t} - x_i^* \rangle. \quad (21)
\end{aligned}
$$

Since $x^*$ is a Nash equilibrium of the convex game, we have that $\langle \nabla_i \mathcal{C}_i(x^*), x_{i,t} - x_i^* \rangle \geq 0$, $i = 1, \ldots, N$. Summing the inequality (21) over $i = 1, \ldots, N$, we get that

$$
\begin{aligned}
\mathbb{E}\|x_{t+1} - x^*\|^2 &\leq \mathbb{E}\|x_t - x^*\|^2 + \eta_t^2 \sum_i \|\nabla_i \mathcal{C}_i(x_t)\|^2 + \eta_t^2 N\sigma^2 - 2\eta_t \sum_i \langle \nabla_i \mathcal{C}_i(x_t), x_{i,t} - x_i^* \rangle \\
&\leq \mathbb{E}\|x_t - x^*\|^2 + \eta_t^2 N B^2 + \eta_t^2 N\sigma^2 - 2\eta_t \sum_i \langle \nabla_i \mathcal{C}_i(x_t) - \nabla_i \mathcal{C}_i(x^*), x_{i,t} - x_i^* \rangle \\
&\leq (1 - 2\eta_t m)\mathbb{E}\|x_t - x^*\|^2 + \eta_t^2 N B^2 + \eta_t^2 N\sigma^2 \\
&\leq (1 - \frac{2}{t}) \|x_t - x^*\|^2 + \frac{N(B^2 + \sigma^2)}{m^2 t^2},
\end{aligned}
$$

where the second inequality follows from Assumption 2 and the fact that $x^*$ is a Nash equilibrium, the third inequality follows from the strong monotonicity of the game (1), and the last inequality is obtained by substituting $\eta_t = \frac{1}{mt}$. Then, using Lemma 5, we obtain

$$
\mathbb{E}\|x_T - x^*\|^2 \leq \frac{N(B^2 + \sigma^2)}{m^2 T} + \mathcal{O}\left(\frac{1}{T}\right),
$$

which completes the proof.

### 8.4 PROOF OF LEMMA 3:

In the analysis that follows, the expectations are taken w.r.t $w_i \sim \mathbb{B}_i$ and $u_{-i} \sim \mathbb{S}_{-i}$. Since $\mathcal{C}_i$ is bounded and with finite support, by Lebesgue's dominated convergence theorem (Chapter 4 in Royden & Fitzpatrick (1988)), we can interchange the order of integration and differentiation. From the definition of $\mathcal{C}_i^\delta(x)$, it follows that

$$
\begin{aligned}
\nabla_i \mathcal{C}_i(x) - \nabla_i \mathcal{C}_i^\delta(x) &= \nabla_i \mathcal{C}_i(x) - \nabla_i \mathbb{E}[\mathcal{C}_i(x_i + \delta w_i, x_{-i} + \delta u_{-i})] \\
&= \mathbb{E}\big[\nabla_i \mathcal{C}_i(x) - \nabla_i \mathcal{C}_i(x_i + \delta w_i, x_{-i} + \delta u_{-i})\big] \\
&\leq \mathbb{E}\big[L_1 \delta \|(w_i, u_{-i})\|\big] \\
&\leq L_1 \delta \sqrt{N}, \quad (22)
\end{aligned}
$$

where the first inequality follows from the Lipschitz continuous property of the function $\nabla_i \mathcal{C}_i(x)$ with respect to $x$. Recall that $\mathcal{A}^\delta$ denotes the set of Nash Equilibria of the smoothed game with

losses $\mathcal{C}_i^\delta(x)$. Take two arbitrary points $y_\delta^*, z_\delta^* \in \mathcal{A}^\delta$. Then, we have that

$$\sum_i \langle \nabla_i \mathcal{C}_i^\delta(y_\delta^*) - \nabla_i \mathcal{C}_i^\delta(z_\delta^*), y_{\delta_i}^* - z_{\delta_i}^* \rangle$$

$$= \sum_i \langle \nabla_i \mathcal{C}_i(y_\delta^*) - \nabla_i \mathcal{C}_i(z_\delta^*), y_{\delta_i}^* - z_{\delta_i}^* \rangle + \sum_i \langle \nabla_i \mathcal{C}_i^\delta(y_\delta^*) - \nabla_i \mathcal{C}_i(y_\delta^*) - \nabla_i \mathcal{C}_i^\delta(z_\delta^*) + \nabla_i \mathcal{C}_i(z_\delta^*), y_{\delta_i}^* - z_{\delta_i}^* \rangle$$

$$\geq \sum_i \langle \nabla_i \mathcal{C}_i(y_\delta^*) - \nabla_i \mathcal{C}_i(z_\delta^*), y_{\delta_i}^* - z_{\delta_i}^* \rangle - \sum_i 2L_1 \delta \sqrt{N} \left\| y_{\delta_i}^* - z_{\delta_i}^* \right\|$$

$$\geq m \left\| y_\delta^* - z_\delta^* \right\|^2 - 2L_1 \delta N \left\| y_\delta^* - z_\delta^* \right\|, \tag{23}$$

where the first inequality follows from the inequality (22) and the second inequality is derived using the strong monotonicity condition (2) and the fact that $(\sum_i \left\| y_{\delta_i}^* - z_{\delta_i}^* \right\|)^2 \leq N \sum_i \left\| y_{\delta_i}^* - z_{\delta_i}^* \right\|^2 = N \left\| y_\delta^* - z_\delta^* \right\|^2$.

Since $y_\delta^*, z_\delta^*$ are Nash Equilibria of the smoothed game, for $\forall x_i \in (1-\delta)\mathcal{X}_i$, we have that $\langle \nabla_i \mathcal{C}_i^\delta(y_\delta^*), x_i - y_{\delta_i}^* \rangle \geq 0$ and $\langle \nabla_i \mathcal{C}_i^\delta(z_\delta^*), x_i - z_{\delta_i}^* \rangle \geq 0$. Replacing $x_i$ in the above two inequalities with $z_{\delta_i}^*$ and $y_{\delta_i}^*$, respectively, we obtain that

$$\langle \nabla_i \mathcal{C}_i^\delta(y_\delta^*), z_{\delta_i}^* - y_{\delta_i}^* \rangle \geq 0, \quad \langle \nabla_i \mathcal{C}_i^\delta(z_\delta^*), y_{\delta_i}^* - z_{\delta_i}^* \rangle \geq 0.$$

Adding these two inequalities together gives that

$$\langle \nabla_i \mathcal{C}_i^\delta(y_\delta^*) - \nabla_i \mathcal{C}_i^\delta(z_\delta^*), y_{\delta_i}^* - z_{\delta_i}^* \rangle \leq 0. \tag{24}$$

Combining inequalities (23) and (24), we get that $m \left\| y_\delta^* - z_\delta^* \right\|^2 - 2L_1 \delta N \left\| y_\delta^* - z_\delta^* \right\| \leq 0$. Rearranging the terms in this inequality completes the proof.

## 8.5 Proof of Lemma 4:

Let $\bar{x}^*$ be the Nash equilibrium of the game defined by the losses $\mathcal{C}_i$ over the set $(1-\delta)\mathcal{X}$, and recall that $x^*$ is the Nash equilibrium of the original game with losses $\mathcal{C}_i$ over the set $\mathcal{X}$ and $x^{\delta,j}$ is one Nash equilibrium of the smoothed game with losses $\mathcal{C}_i^\delta$ over the set $(1-\delta)\mathcal{X}$. Since the losses $\mathcal{C}_i$ are strongly monotone, the Nash equilibrium $\bar{x}^*$ is unique and well-defined.

By the triangle inequality, we have that $\left\| x^* - x^{\delta,j} \right\| \leq \left\| x^* - \bar{x}^* \right\| + \left\| x^{\delta,j} - \bar{x}^* \right\|$. We first bound $\left\| x^* - \bar{x}^* \right\|$. This bound has already been derived by Drusvyatskiy & Ratliff (2021), and takes the form

$$\left\| x^* - \bar{x}^* \right\| \leq \delta \left( 1 + \frac{L_1 \sqrt{N}}{m} \right) \left\| x^* \right\|. \tag{25}$$

Next we focus on the term $\left\| x^{\delta,j} - \bar{x}^* \right\|$. The convexity of the game guarantees that the Nash Equilibria satisfy the following property:

$$\langle \nabla_i \mathcal{C}_i(\bar{x}^*), x_i - \bar{x}_i^* \rangle \geq 0, \quad \forall x_i \in (1-\delta)\mathcal{X}_i,$$
$$\langle \nabla_i \mathcal{C}_i^\delta(x^{\delta,j}), x_i - x_i^{\delta,j} \rangle \geq 0, \quad \forall x_i \in (1-\delta)\mathcal{X}_i. \tag{26}$$

Replacing $x_i$ in the above two inequalities in (26) with $x_i^{\delta,j}$ and $\bar{x}_i^*$, respectively, and summing the two inequalities, we have that

$$\langle \nabla_i \mathcal{C}_i(\bar{x}^*) - \nabla_i \mathcal{C}_i^\delta(x^{\delta,j}), x_i^{\delta,j} - \bar{x}_i^* \rangle \geq 0. \tag{27}$$

Combining (27) with the strong monotonicity condition (2), we have that

$$m \left\| x^{\delta,j} - \bar{x}^* \right\|^2 \leq \sum_i \langle \nabla_i \mathcal{C}_i(x^{\delta,j}) - \nabla_i \mathcal{C}_i(\bar{x}^*), x_i^{\delta,j} - \bar{x}_i^* \rangle$$

$$\leq \sum_i \langle \nabla_i \mathcal{C}_i(x^{\delta,j}) - \nabla_i \mathcal{C}_i^\delta(x^{\delta,j}), x_i^{\delta,j} - \bar{x}_i^* \rangle$$

$$\leq \sum_i L_1 \delta \sqrt{N} \left\| x_i^{\delta,j} - \bar{x}_i^* \right\|$$

$$\leq L_1 \delta N \left\| x^{\delta,j} - \bar{x}^* \right\|,$$

where the first inequality is due to the strong monotonicity condition (2), the second inequality follows from (27) and the third inequality follows from (22). Therefore, for any Nash equilibrium $x^{\delta,j} \in \mathcal{A}^\delta$, we have that

$$\left\| x^{\delta,j} - \bar{x}^* \right\| \leq \frac{L_1 \delta N}{m}. \tag{28}$$

Combining (28) with (25), we have that $\left\| x^* - x^{\delta,j} \right\| \leq \left\| x^* - \bar{x}^* \right\| + \left\| x^{\delta,j} - \bar{x}^* \right\| \leq \delta \left(1 + \frac{L_1 \sqrt{N}}{m}\right) \|x^*\| + \frac{L_1 \delta N}{m}$, which completes the proof.

## 8.6 PROOF OF THEOREM 3:

Recall that the Nash Equilibria of the smoothed game with losses $\mathcal{C}_i^\delta$ over the set $(1-\delta)\mathcal{X}$ are $x^{\delta,j}$. From the update equation (7), for any $x^{\delta,j} \in \mathcal{A}^\delta$, we have that

$$\begin{aligned}
\left\| x_{i,t+1} - x_i^{\delta,j} \right\|^2 &= \left\| \mathcal{P}_{(1-\delta)\mathcal{X}_i}(x_{i,t} - \eta_t g_{i,t}) - x_i^{\delta,j} \right\|^2 \\
&\leq \left\| x_{i,t} - x_i^{\delta,j} - \eta_t g_{i,t} \right\|^2 \\
&\leq \left\| x_{i,t} - x_i^{\delta,j} \right\|^2 + \eta_t^2 \|g_{i,t}\|^2 - 2\eta_t \langle g_{i,t}, x_{i,t} - x_i^{\delta,j} \rangle,
\end{aligned}$$

where the first inequality holds since $x_i^{\delta,j} \in (1-\delta)\mathcal{X}_i$. Taking expectations with respect to $u_{i,t}$ of both sides of the above inequality, we have that

$$\mathbb{E}\left\| x_{i,t+1} - x_i^{\delta,j} \right\|^2 \leq \mathbb{E}\left\| x_{i,t} - x_i^{\delta,j} \right\|^2 + \frac{\eta_t^2 d_i^2 U^2}{\delta^2} - 2\eta_t \langle \nabla_i \mathcal{C}_i^\delta(x_t), x_{i,t} - x_i^{\delta,j} \rangle. \tag{29}$$

Since $x^{\delta,j}$ is a Nash equilibrium of the smoothed game, it satisfies $\langle \nabla_i \mathcal{C}_i^\delta(x^{\delta,j}), x_i - x_i^{\delta,j} \rangle \geq 0, \forall x_i \in (1-\delta)\mathcal{X}_i$. Summing the inequality (29) over $i = 1, \ldots, N$, we get

$$\begin{aligned}
\mathbb{E}\left\| x_{t+1} - x^{\delta,j} \right\|^2 &\leq \mathbb{E}\left\| x_t - x^{\delta,j} \right\|^2 + \frac{\eta_t^2 d_i^2 U^2 N}{\delta^2} - 2\eta_t \sum_i \langle \nabla_i \mathcal{C}_i^\delta(x_t) - \nabla_i \mathcal{C}_i^\delta(x^{\delta,j}), x_{i,t} - x_i^{\delta,j} \rangle \\
&\leq \mathbb{E}\left\| x_t - x^{\delta,j} \right\|^2 + \frac{\eta_t^2 d_i^2 U^2 N}{\delta^2} - 2\eta_t \left( m \left\| x_t - x^{\delta,j} \right\|^2 - 2L_1 \delta N \left\| x_t - x^{\delta,j} \right\| \right) \\
&\leq (1 - 2\eta_t m)\mathbb{E}\left\| x_t - x^{\delta,j} \right\|^2 + \frac{\eta_t^2 d_i^2 U^2 N}{\delta^2} + 4\eta_t \delta L_1 ND, \tag{30}
\end{aligned}$$

where the second inequality can be obtained using similar techniques as in (23). Substituting $\eta_t = \frac{1}{mt}$ into (30), we get that

$$\begin{aligned}
\mathbb{E}\left\| x_{t+1} - x^{\delta,j} \right\|^2 &\leq (1 - \frac{2}{t})\mathbb{E}\left\| x_t - x^{\delta,j} \right\|^2 + \frac{d_i^2 U^2 N}{4m^2 t^2 \delta^2} + \frac{2L_1 ND\delta}{mt} \\
&\leq (1 - \frac{2}{t})\mathbb{E}\left\| x_t - x^{\delta,j} \right\|^2 + \frac{E_1}{t^2 \delta^2} + \frac{E_2 \delta}{t}, \tag{31}
\end{aligned}$$

where $E_1 = \frac{d_i^2 U^2 N}{4m^2} + \frac{2L_1 ND}{m}$, $E_2 = \frac{2L_1 ND}{m}$. Using induction, it can be verified that for $\forall t \geq 1$, there exists a constant $A_0 > 0$ such that

$$\mathbb{E}\left\| x_t - x^{\delta,j} \right\|^2 \leq \max \left\{ \frac{E_1}{t\delta^2} + E_2 \delta, A_0 \left( \frac{1}{t\delta^2} + \delta \right) \right\}. \tag{32}$$

Replacing $t$ with $T$ and setting $\delta = T^{-\frac{1}{3}}$ in (32), we get

$$\mathbb{E}\left\| x_T - x^{\delta,j} \right\|^2 \leq \max \left\{ \frac{E_1}{T\delta^2} + E_2 \delta, A_0 \left( \frac{1}{T\delta^2} + \delta \right) \right\} \leq 2A_1 T^{-\frac{1}{3}}, \tag{33}$$

where $A_1 := \max\{E_1, E_2, A_0\}$. Combining (33) with Lemma 4, we have

$$\begin{aligned}
\mathbb{E}\left\| x_T - x^* \right\|^2 &\leq 2\mathbb{E}\left\| x_T - x^{\delta,j} \right\|^2 + 2\left\| x^{\delta,j} - x^* \right\|^2 \\
&\leq 4A_1 T^{-\frac{1}{3}} + \frac{2\left( \left(1 + \frac{L_1 \sqrt{N}}{m}\right) \|x^*\| + \frac{L_1 N}{m} \right)^2}{T^{\frac{2}{3}}} \\
&= \mathcal{O}\left( T^{-\frac{1}{3}} \right), \tag{34}
\end{aligned}$$

which completes the proof.

## 8.7 PROOF OF THEOREM 4:

From the convexity of the loss function $\mathcal{C}_i$ and the update rule (10), we have that

$$\langle \nabla_i \mathcal{C}_i(x_{i,t+1}, x_{-i,t}), x_i - x_{i,t+1} \rangle \geq 0, \quad \forall x_i \in \mathcal{X}_i. \tag{35}$$

Since the game (1) is strongly monotone, we also have that for all $x_i \in \mathcal{X}_i$,

$$\langle \nabla_i \mathcal{C}_i(x_i, x_{-i,t}) - \nabla_i \mathcal{C}_i(x_{i,t+1}, x_{-i,t}), x_i - x_{i,t+1} \rangle \geq m \|x_i - x_{i,t+1}\|^2. \tag{36}$$

Replacing $x_i$ with $x_i^*$ in (36) and combining with (35), we get

$$m \|x_i^* - x_{i,t+1}\|^2 \leq \langle \nabla_i \mathcal{C}_i(x_i^*, x_{-i,t}) - \nabla_i \mathcal{C}_i(x_{i,t+1}, x_{-i,t}), x_i^* - x_{i,t+1} \rangle \leq \langle \nabla_i \mathcal{C}_i(x_i^*, x_{-i,t}), x_i^* - x_{i,t+1} \rangle. \tag{37}$$

Summing the inequality (37) over $i = 1, \ldots, N$, it follows that

$$
\begin{aligned}
\|x_{t+1} - x^*\|^2 &\leq \frac{1}{m} \sum_i \langle \nabla_i \mathcal{C}_i(x_i^*, x_{-i,t}), x_i^* - x_{i,t+1} \rangle \\
&\leq \frac{1}{m} \sum_i \langle \nabla_i \mathcal{C}_i(x_i^*, x_{-i,t}) - \nabla_i \mathcal{C}_i(x^*), x_i^* - x_{i,t+1} \rangle \\
&\leq \frac{1}{m} \sum_i L \|x_{-i,t} - x_{-i}^*\| \|x_i^* - x_{i,t+1}\| \\
&\leq \frac{L}{m} \sum_i \frac{1}{2\lambda} \|x_{-i,t} - x_{-i}^*\|^2 + \frac{\lambda}{2} \|x_{i,t+1} - x_i^*\|^2 \\
&\leq \frac{L}{m} \left( \frac{N-1}{2\lambda} \|x_t - x^*\|^2 + \frac{\lambda}{2} \|x_{t+1} - x^*\|^2 \right),
\end{aligned}
\tag{38}
$$

where the second inequality follows from the Nash equilibrium condition $\langle \nabla_i \mathcal{C}_i(x^*), x_i - x_i^* \rangle \geq 0$, for $\forall x \in \mathcal{X}$, the third inequality follows from the Lipschitz continuous property of the function $\mathcal{C}_i$ in $x_{-i}$, and the second to last inequality is due to the fact that $ab \leq \frac{1}{2\lambda} a^2 + \frac{\lambda}{2} b^2$ for any $\lambda > 0$. Setting $\lambda = \frac{m}{L}$ and rearranging the terms in (38), we obtain that

$$\|x_{t+1} - x^*\|^2 \leq \frac{L^2(N-1)}{m^2} \|x_t - x^*\|^2. \tag{39}$$

Applying inequality (39) iteratively for $t = 1, \ldots, T$, we have that $\|x_T - x^*\|^2 \leq \left( \frac{L^2(N-1)}{m^2} \right)^T \|x_0 - x^*\|^2$, which completes the proof.

## 8.8 BR ALGORITHM IS NO-REGRET LEARNING

**Theorem 5.** *Suppose that the game* (1) *is $m$-strongly monotone with parameter $m > L\sqrt{N-1}$. Then the BR Algorithm achieves no-regret learning, specifically*

$$\text{Reg}_i = \sum_{t=1}^{T} \mathcal{C}_i(x_t) - \min_{x_i} \sum_{t=1}^{T} \mathcal{C}_i(x_i, x_{-i,t}) = \mathcal{O}(\sqrt{T}). \tag{40}$$

*Proof.* Let $y_i^* := \operatorname{argmin}_{x_i} \sum_{t=1}^{T} \mathcal{C}_i(x_i, x_{-i,t})$. Recalling that $x_{i,t+1} = \operatorname{argmin}_{x_i} \mathcal{C}_i(x_i, x_{-i,t})$, we have that $\langle \nabla_i \mathcal{C}_i(x_{i,t+1}, x_{-i,t}), y_i^* - x_{i,t+1} \rangle \geq 0$. From the definition of regret in (3), we have that

$$
\begin{aligned}
\text{Reg}_i &= \sum_{t=1}^{T} \left( \mathcal{C}_i(x_t) - \mathcal{C}_i(y_i^*, x_{-i,t}) \right) \\
&= \sum_{t=1}^{T} \left( \mathcal{C}_i(x_t) - \mathcal{C}_i(x_{i,t+1}, x_{-i,t}) + \mathcal{C}_i(x_{i,t+1}, x_{-i,t}) - \mathcal{C}_i(y_i^*, x_{-i,t}) \right) \\
&\leq \sum_{t=1}^{T} \left( \mathcal{C}_i(x_t) - \mathcal{C}_i(x_{i,t+1}, x_{-i,t}) \right) + \sum_{t=1}^{T} \langle \nabla_i \mathcal{C}_i(x_{i,t+1}, x_{-i,t}), x_{i,t+1} - y_i^* \rangle \\
&\leq \sum_{t=1}^{T} \left( \mathcal{C}_i(x_t) - \mathcal{C}_i(x_{i,t+1}, x_{-i,t}) \right),
\end{aligned}
$$

where the first inequality is due to the convexity of the loss function $\mathcal{C}_i(x)$ with respect to $x_i$ and the second inequality follows from the necessary condition of optimality. Then, it follows that

$$
\begin{aligned}
\text{Reg}_i &\leq \sum_{t=1}^{T} \Big( \mathcal{C}_i(x_t) - \mathcal{C}_i(x_{t+1}) + \mathcal{C}_i(x_{t+1}) - \mathcal{C}_i(x_{i,t+1}, x_{-i,t}) \Big) \\
&\leq \mathcal{C}_i(x_1) + \sum_{t=1}^{T} \Big( \mathcal{C}_i(x_{t+1}) - \mathcal{C}_i(x_{i,t+1}, x_{-i,t}) \Big) \\
&\leq U + L_0 \sum_{t=1}^{T} \| x_{-i,t+1} - x_{-i,t} \|,
\end{aligned}
\tag{41}
$$

where the last inequality follows from the Lipschitz continuous property of the function $\mathcal{C}_i$ in $x$. Summing the inequality (41) over $i = 1, \ldots, N$, we have that

$$
\sum_{i=1}^{N} \text{Reg}_i \leq NU + L_0 \sum_{i=1}^{N} \sum_{t=1}^{T} \| x_{-i,t+1} - x_{-i,t} \| \leq NU + L_0 \sqrt{N(N-1)} \sum_{t=1}^{T} \| x_{t+1} - x_t \|,
\tag{42}
$$

where the last inequality follows from the fact that $(\sum_i \| x_{-i,t+1} - x_{-i,t} \|)^2 \leq N \sum_i \| x_{-i,t+1} - x_{-i,t} \|^2 \leq N(N-1) \| x_{t+1} - x_t \|^2$. From the inequality (39) in the proof of Theorem 4, when $m > L\sqrt{N-1}$, we have that $\| x_{t+1} - x^* \|^2 \leq \rho \| x_t - x^* \|^2$. Using this result, we can bound the term $\| x_{t+1} - x_t \|^2$ as follows:

$$
\begin{aligned}
\| x_{t+1} - x_t \|^2 &= \| x_{t+1} - x^* + x^* - x_t \|^2 \\
&= \| x_{t+1} - x^* \|^2 + \| x_t - x^* \|^2 - 2 \| x_{t+1} - x^* \| \| x_t - x^* \| \\
&\leq \| x_{t+1} - x^* \|^2 + \| x_t - x^* \|^2 - \frac{2}{\sqrt{\rho}} \| x_{t+1} - x^* \|^2 \\
&\leq \| x_t - x^* \|^2 - \| x_{t+1} - x^* \|^2,
\end{aligned}
\tag{43}
$$

where the last inequality holds since $\rho < 1$. Then, substituting the inequality (43) into the inequality (42), we get that

$$
\begin{aligned}
\sum_{i=1}^{N} \text{Reg}_i &\leq NU + L_0 \sqrt{N(N-1)} \sum_{t=1}^{T} \sqrt{\| x_t - x^* \|^2 - \| x_{t+1} - x^* \|^2} \\
&\leq NU + L_0 \sqrt{N(N-1)T} \| x_1 - x^* \| \\
&\leq NU + L_0 D \sqrt{N(N-1)T},
\end{aligned}
\tag{44}
$$

where the second inequality holds since $(\sum_{t=1}^{T} a_t)^2 \leq T \sum_{t=1}^{T} a_t^2$. Using the fact that $\text{Reg}_i \leq \sum_{i=1}^{N} \text{Reg}_i$ completes the proof. $\qquad \square$

From equation (41) in the convergence proof of the best response algorithm, we have that $\text{Reg}_i \leq \sum_{t=1}^{T} \Big( \mathcal{C}_i(x_t) - \mathcal{C}_i(x_{i,t+1}, x_{-i,t}) \Big) = \sum_{t=1}^{T} \mathcal{C}_i(x_t) - \sum_{t=1}^{T} \min_{x_i \in \mathcal{X}_i} \mathcal{C}_i(x_i, x_{-i,t}) = \mathcal{O}(\sqrt{T})$, which indicates that the dynamic regret of agent $i$, i.e., $\sum_{t=1}^{T} \mathcal{C}_i(x_t) - \sum_{t=1}^{T} \min_{x_i \in \mathcal{X}_i} \mathcal{C}_i(x_i, x_{-i,t})$, is of order $\mathcal{O}(\sqrt{T})$. Note that this term captures the dynamic regret induced by the variation of other agents' actions.

## 8.9 ADDITIONAL EXPERIMENTS: RETAILER PRICING GAME

Consider a market with two retailers and four products. Each retailer is responsible for selling different products and making pricing decisions for their own products at each episode $t$. Suppose that Retailer 1 sells the $j$-th product for $j = 1, 2$, while Retailer 2 sells the $j$-th product for $j = 3, 4$. Retailers 1 and 2 make pricing decisions $x_1^t = (p_1^t, p_2^t)$ and $x_2^t = (p_3^t, p_4^t)$, respectively, where $x_i^t$ denotes the decision of Retailer $i$, and $p_j^t$ denotes the price of the $j$-th product at episode $t$.

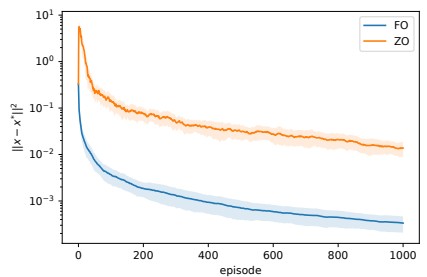 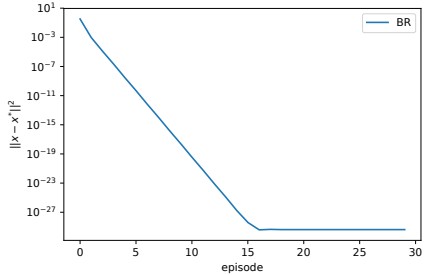

(a) Error to Nash equilibrium of the first-order method (FO) and the zeroth-order method (ZO).

(b) Error to Nash equilibrium of the best response algorithm (BR).

Figure 4: Retailer pricing game when $m > L\sqrt{N-1}$.

All products are substitutes or complements of each other and their pricing decisions influence the demand of other products. The demand of the $j$-th product is modeled as $D_j(x) = \sum_{k=1}^4 A_{jk} p_k^t + b_j$, where $A_{jk}$, $b_j > 0$ are constants. Note that $A_{jk} > 0$ if the $k$-th product is a substitute for the $j$-th product, $A_{jk} < 0$ if the $k$-th product is a complement of the $j$-th product, and $A_{jj} < 0$ for $j = 1, 2, 3, 4$. The goal of each retailer is to minimize their own loss, where $C_1(x) = -(p_1 D_1(x) + p_2 D_2(x))$ and $C_2(x) = -(p_3 D_3(x) + p_4 D_4(x))$. The parameters $A_{jk}$ are given as

$$A_{1\leq j,k\leq 4} = \begin{bmatrix} -2 & -0.5 & 0.1 & -0.1 \\ -0.5 & -2 & -0.4 & 0.1 \\ 0.1 & -0.4 & -2 & -0.2 \\ -0.1 & 0.1 & -0.2 & -2 \end{bmatrix}.$$

Using these parameters, we get that $m = 2.82$, $L = 0.3$, and, therefore, $m > L\sqrt{N-1}$. As a result, the sufficient conditions are satisfied for all three algorithms. The results are shown in Figure 4, where the solid lines and shaded areas denote the average and standard deviation over 60 runs. Figure 4 shows that all three algorithms converge to the Nash equilibrium.

## 8.10    ADDITIONAL EXPERIMENTS: KELLY AUCTION

Consider a service provider (providing, e.g., bandwidth, space on a website, etc.) with two bidders $N = 2$ that place monetary bids $x_i \in [0, D]$ for the utilization of the resource. Each bidder receives some unit of the resource which is proportional to their own bid, i.e., the $i$-th bidder gets $\rho_i = \frac{x_i}{d + \sum_{i=1}^N x_i}$ units of resource, where $d \geq 0$ is the entry barrier for bidding on it. We model the loss of each bidder as $u_i(x) = -(g_i \rho_i - x_i)$, where $g_i$ is the bidder's marginal gain from obtaining a unit of resources. We set $g_i = 0.8$, $\forall i = 1, 2$ and $d = 0.2$. Figure 5 presents simulation results for this example. We see that the first-order and zeroth-order methods converge to the Nash equilibrium. Note also that, in this example, $m < L\sqrt{N-1}$ always holds regardless of the choice of parameters. Therefore, the sufficient condition for the best response algorithm is not satisfied. Nevertheless, as seen in Figure 5(b), the best response algorithm still converges to the Nash equilibrium, which validates our theoretical analysis that this condition is only sufficient and not necessary.

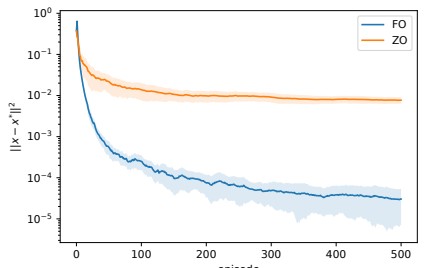

(a) Error to Nash equilibrium of the first-order method (FO) and the zeroth-order method (ZO).

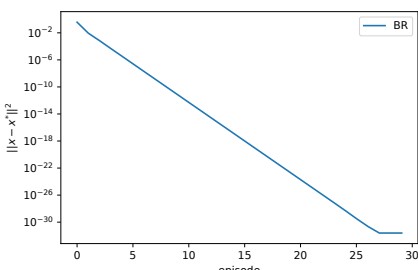

(b) Error to Nash equilibrium of the best response algorithm (BR).

Figure 5: Kelly auction game.

