# OpenReview forum: "No-Regret Learning in Strongly Monotone Games Converges to a Nash Equilibrium"
_ICLR.cc/2023/Conference — Submitted to ICLR 2023_

### Official Review · Reviewer_k8bM · 2022-10-25

**Confidence:** 4
**Clarity, Quality, Novelty And Reproducibility:** (Please see above)
**Correctness:** 4
**Technical Novelty And Significance:** 2
**Empirical Novelty And Significance:** 2
**Recommendation:** 6

**Strength And Weaknesses:**

*The paper visibly exceeds the allowed margins by a significant amount, so it is not clear whether it can proceed with the review process. Unfortunately, due to understandable fairness reasons, I believe there is a chance that the paper might need to be rejected; I will let the area chairs deliberate on that. In what follows, I will focus on the content of the paper under the assumption that the paper will be allowed, hoping that even if the paper is rejected the authors will still find the comments useful.*

---

Overall, I have a borderline opinion of the paper, more positive than negative. I didn't find any of the results especially surprising, but I find the paper well organized, and the bundling of topic well executed, with only a few reasons to be unhappy.

1. What I find the most prominent weakness of the paper is the setting of strongly monotone games, which I find pretty restrictive, though of course I would be happy to be proven wrong by the authors. In particular, I'd be interested in knowing games of interest with more than two players that satisfy strong monotonicity. All the examples considered by the authors seem to have two players.

2. (Adding more to point 1). I might be being silly, but the important conditions on $m$ in Theorem 1 and Theorem 4 require that the ratio $m/L$ be $m/L > O(1)\sqrt{N-1}$. The reason why I find this a bit surprising is that $L$ is a Lipschitz constant on the gradients of the losses (fundamentally an **upper** bound on the variation), and $m$ is a strong convexity modulus for the gradients of the losses (fundamentally a **lower** bound on the variation). Hence, at least on some intuitive level one would imagine that usually $m/L < 1$, yet the conditions in the theorem require $m/L \ge \sqrt{N-1} \ge 1$ for $N \ge 2$. Maybe this is what the authors were trying to get at with Remark 1, but unless I'm missing something, this to me further reduces the applicability and appeal of the results in the paper in a way that Remark 1 was maybe too shy to clearly point out. Please let me know if I missed anything.

3. Another area where the paper might have easily been improved was in giving high probability bounds rather than only in-expectation (I imagine that that would follow easily from concentration inequalities, but please let me know if I missed something).

4. I found the description of the bandit setting, so focused on whether players are comfortable or willing to reveal actions, a bit unusual. I think perhaps the motivation for the bandit setting would have been better found in whether or not the players know their payoff matrix, or whether instead they need to learn that as well.

Overall, my biggest reservations lie around my points 1 and 2 above; if those were resolved, my score for the paper would easily increase.

Nits / Typos:
- At the end of page 3, there is a double comma ",,"

**Summary Of The Paper:**

The paper studies the convergence of no-regret dynamics in smooth strongly monotone games with sufficiently large strong convexity modulus.

**Summary Of The Review:**

Ignoring the violation of the formatting guidelines, I find that the the reasons to like the paper outweigh the reasons to dislike it. However, I am slightly unconvinced about the applicability of the paper, and that weighs on my score.

---

> ### Author Response · Authors · 2022-11-15
> **Thank you for your review. See answers below.**
>
> Thank you for observing the formatting error. This was a careless mistake on our end and we sincerely apologize for it. We have revised our paper according to the ICLR formatting rules. We would greatly appreciate it if this formatting error did not result in rejection of this paper, and instead this work was reviewed for its contribution. Of course, we will respect the final decision of the reviewers and the area chair, whichever it is. We thank the reviewer for giving us valuable feedback and comments even in this case. Please find the responses below.
>
> 1. Strongly monotone games: The strong monotonicity assumption can be restrictive, but there are still many games that satisfy this assumption, e.g., Cournot games, retailer pricing games, Kelly auctions. In fact, strongly monotone games are widely investigated in the literature (see "Bravo Mario, Bandit learning in concave n-person games, 2018”; “Tianyi Lin, Optimal no-regret learning in strongly monotone games with bandit feedback, 2021"). Moreover, there is no strict limitation on the numbers of agents in strongly monotone games (see “ Rosen J. B., Existence and uniqueness of equilibrium points for concave n-person games, 1965”). There may be N players in strongly monotone games for a large N. In our simulations, we also consider a Cournot game with multiple players (N=5) and the experimental results are illustrated in Fig. 3. For simplicity, for the Retailer pricing game and Kelly auction game we present results for two agents but we can handle any number of agents, as long as the condition is satisfied.
>
> 2. Relation between L and m and $L_1$: First we want to recall their definitions: L is the Lipschitz constant of $\nabla_i C_i(x_i,x_{-i})$ w.r.t. $x_{-i}$, $L_1$ is the Lipschitz constant of $\nabla_i C_i(x)$ w.r.t. $x$ (fundamentally an upper bound), $m$ is the strongly monotone parameter (fundamentally a lower bound). From these definitions, it always holds that $L\leq L_1$. We agree the reviewer’s intuition is correct: the strongly convex parameter is always less than the Lipschitz continuous parameter, so it always holds that $m\leq L_1$. However, it is possible to have $m\leq L$, e.g., in the extreme case that $C_i$ only depends on agent i’s own action $x_i$ we have $L=0$. As mentioned in Remark 1, the parameter $L$ can be interpreted as a bound on the maximum influence of the other agents’ actions, and there is no relation between $L$ and $L_1$. Therefore, there exist classes of games in which $m\geq L\sqrt{N-1}$. Thanks for pointing out this, and we also give more explanation in the revised paper.
>
> 3. High probability bounds: Thank you for the great suggestion. Obtaining high probability results is definitely possible. This is left for our future work.
>
> 4. Motivation of Bandit setting: There are many applications where the agents can only access the function evaluations rather than the true gradient. For example, gradient feedback $\nabla_i C_i(x_i,x_{-i})$ is not accessible if agent $i$ does not know the payoff gradient function, or cannot observe other agents' actions $x_{-i}$. Besides, in risk-averse games in which agents aim to minimize the Conditional Value at Risk (CVaR) of their cost functions, we can only evaluate CVaR from samples and therefore cannot have closed-form gradients.

---

### Official Review · Reviewer_yciX · 2022-10-26

**Confidence:** 2
**Correctness:** 4
**Technical Novelty And Significance:** 3
**Empirical Novelty And Significance:** 2
**Recommendation:** 6

**Clarity, Quality, Novelty And Reproducibility:**


The paper could be organized better -- I found it difficult to parse the main contributions. An attempt to classify games and algorithms in this manner is original. The experiments could have been higher quality -- it was hard to extract the relevant points as many details were presented here.

**Strength And Weaknesses:**

# Strengths
- Finding Nash equilibria is an important problem in game theory. It is interesting to identify classes of games for which algorithms of a certain type will always converge to a Nash equilibrium. However, the m-strongly monotone games are a special type of convex game, in which all loss functions for each agent are convex. This seems to be a strong assumption on the game: for these games, only one Nash equilibrium exists.
- The experimental analysis of the convergence of various algorithms on specific examples of Cournot games and retail pricing games is nice to see. The authors show that when their assumptions are not satisfied, some of the algorithms may cycle and never converge.

# Weaknesses
- Strong assumptions needed on the loss functions. In addition to m-strongly monotone, convex loss functions, Assumption 1 & 2 on Lipschitz continuity are needed. However, the authors identify specific applications in which these assumptions do hold, including Cournot games and retail pricing games.
- Theorem 1 appears to be the main result of the paper. However, there is very little discussion of the Theorem or its proof in the main text. Much space is devoted to the gradient descent algorithms.
- It would be nice if the authors identified specific algorithms, already extant in the literature, that fall into their framework but were not known to converge to Nash equilbrium. However, I did not see any discussion of this kind of result. Such algorithms would show the utility of the analysis.

**Summary Of The Paper:**

This paper analyzes classes of online games for which classes of algorithms will always converge to a Nash equilibrium. The main result is: for m-strongly monotone games, any no-regret algorithm will converge to a Nash equilibrium, if m satisfies certain conditions. The paper also considers other types of algorithms: first-order and zeroth-order gradient descent algorithms. Experimental results validate the theoretical results on a few examples of strongly monotone online games. Also, best response algorithms are considered.

**Summary Of The Review:**


I think this is a borderline paper, tending to accept, based upon my above comments.

---

> ### Author Response · Authors · 2022-11-15
> **Thank you for your review. See answers below.**
>
> Thank you for your valuable feedback and comments. Please find the responses below.
>
> 1. Strongly monotone games: As mentioned in our response to another reviewer, note that Theorem 1 is not restricted to any specific algorithm. Therefore, it is reasonable to expect that such a general result that applies to any no-regret learning algorithm will need some assumptions on the type of games it can handle that may be restrictive. In a sense, we tradeoff generality in the selection of the algorithm for some restriction in the selection of the game. We believe that such an approach is quite novel compared to related literature that typically does the opposite; it restricts the selection of the algorithm and focuses on least restrictive conditions on the game. We also note that the sufficient condition $m>2L\sqrt{N-1}$ is not necessary and, in general, can be conservative. Nevertheless, at the same time, we present a counterexample showing that the best-response algorithm may not converge to a Nash equilibrium when $m<L\sqrt{N-1}$; see Fig. 2 in our experimental results. This shows that the proposed sufficient condition $m>2L\sqrt{N-1}$ in some cases can be tight. We believe this result is interesting and supports the validity of the proposed strong monotonicity assumption (recall again that Theorem 1 holds for any learning algorithm as long as it is no-regret).  We finally note that the strong monotonicity assumption on the class of games and the Lipschitz continuity assumption of the function $\nabla_i C_i(x)$ w.r.t. $x_{-i}$ needed to show Theorem 1 hold in many applications, e.g., Cournot games, retailer pricing games and Kelly auction games.
>
> 2. Assumptions 1 & 2 are strong: First, note that the results in Theorem 1 (the most significant contribution of this paper) and Theorem 4 do not need Assumptions 1 & 2. They only need to assume strongly monotone games and the Lipschitz continuity of the function $\nabla_i C_i(x)$ w.r.t. $x_{-i}$.  The convergence results of gradient descent methods, i.e., Theorem 2 & 3 needs Assumptions 1 & 2. However, Assumptions 1 & 2 on Lipschitz continuity, as mentioned by the reviewer, hold for many types of games, e.g., Cournot games, retail pricing games, Kelly auction games, and are widely used in the literature (see "Bravo Mario, Bandit learning in concave n-person games, 2018”; “Tianyi Lin, Optimal no-regret learning in strongly monotone games with bandit feedback, 2021").
>
> 3. Little discussion of Theorem 1: Theorem 1 is indeed the key result of this paper, and we have added more discussion on when this condition can be satisfied, and some intuitive explanations. Thanks for pointing out.
>
> 4. Identify specific algorithms that fall into framework but not known to converge to Nash equilibrium: One way to use our result is the following. In general, it is easier to show an algorithm is no-regret compared to showing it converges to a Nash equilibrium. There are many algorithms in games that are shown to be no-regret (e.g., ExtraGradient, Optimistic gradient, mirror descent, best-response). Our result says that as long as we can show an algorithm is no-regret and a game satisfies a monotonicity condition, we get Nash equilibrium convergence for free. Therefore, all these no-regret algorithms are able to achieve Nash equilibrium convergence in the class of m-strongly monotone games with $m>2L\sqrt{N-1}$. This positive result made the subsequent convergence analysis meaningful. Of course, with knowledge of specific algorithms (like how it performs updates), it is likely to relax the condition like we do in this paper. For example,  the best response algorithm in this paper is, to the best of our knowledge, new and not known to converge to Nash equilibrium. But our result shows that when $m>L\sqrt{N-1}$, the best-response algorithm will converge to the Nash equilibrium.
>
> 5. Organization: Thanks for the advice. We have re-organized the paper to highlight the key points in this paper.

---

### Official Review · Reviewer_yhnp · 2022-10-27

**Confidence:** 5
**Correctness:** 4
**Technical Novelty And Significance:** 3
**Empirical Novelty And Significance:** 2
**Recommendation:** 3

**Clarity, Quality, Novelty And Reproducibility:**


Clarity & Quality: The proofs are really well-written
Novelty: See above. Additionally, the case of Gradient Ascent in Strongly-monotone games seems in my humble opinion, trivial generalization of convergence rate of GD in strongly convex minimization. Similarly, best response dynamics are trivially-expectedly well fitted in the convergence rate, since ExtraGradient and Optimistic methods aim exactly to approximate the implicit best-response problem. Therefore, the main contribution, besides the connection of no-regret and convergence, only zero-order should be mentioned as a novel part.

Additionally, after some investigation in the recent literature, I noticed the existence of a hardness result
(The Computational Complexity of Multi-player Concave Games and Kakutani Fixed Points
Authors: Christos H. Papadimitriou, Emmanouil-Vasileios Vlatakis-Gkaragkounis, Manolis Zampetakis) which provides a PPAD-ness completeness for a specific class of strongly concave games with 2 players. It would be interesting the authors to include a short paragraph to explain why their algorithm can not resolve in polynomial case the underlying hard case.


Finally, an extra comment, which should be better adjusted as a term, is the usage of the word (online class of games). In the analysis, the players learn about the game via their oracles access in an online way. However, the underlying game remains the same as the time-steps are going forward, Therefore, the characterization ''online'' belongs to the dynamics with which the single game is endowed rather than the underlying game.

**Details Of Ethics Concerns:**

Non-applicable

**Strength And Weaknesses:**

This is a paper that had created to me a very controversial feeling. The majority of the presented results, in some sense it seems to me already known from the literature, especially from the line of Mertikopoulos et al ' works. Additionally I noticed that the authors have not cited and compare their work with the following publication, "Tight last-iterate convergence rates for no-regret learning in multi-player games".

In general the proofs, as machinery, are tremendously used again again last years in the literature.
However, I should mention that in this draft, the proofs are definitely clear.

The real strength of the arguments of this work is the clear connection of no-regret guarantee of an algorithm and its convergence guarantee if the underlying game is strongly-monotone.

While I have worked in the area, I am not sure if such a clear connection has been established before and I would like to see from the discussion and the opinion of the Area Chair, if this is an actual novel contribution.


**Summary Of The Paper:**

This paper aims to understand the convergence rates of no-regret methods in strongly monotone games to Nash equilibria. It examines first-order/zero-order/best-response instantiations proving their no-regret guarantees. The main contribution of this work is to show that the strong-monotone class of games for which no-regret learning leads to a Nash equilibrium can be expanded if only some further information on the learning algorithm is known, like the regret rate.

**Summary Of The Review:**

The authors address this question by providing a sufficient condition for strongly monotone games that guarantees Nash equilibrium convergence in a time average & last-iterate sense (for some cases).  Specifically, they provide relaxed sufficient conditions for first-order and zeroth-order gradient descent algorithms as well as for best response algorithms in which agents choose actions that best respond to other players' actions during the last episode.

---

> ### Author Response · Authors · 2022-11-15
> **Thank you for your review. See answers below.**
>
> Thank you for your valuable feedback and comments. Please find the responses below.
>
> 1. Comparison with [1]: We agree that [1] is closely related to this paper, which proposes an optimistic gradient method (a kind of first-order method) that achieves no-regret learning and tight last-iterate Nash equilibrium convergence in smooth monotone games. We have added the comparison in the Introduction part.
>
> 2. Connection of no-regret learning (regardless of the algorithm) and Nash equilibrium convergence: This is exactly the most significant contribution of this paper as shown in Theorem 1. To the best of our knowledge, there is no work to analyze the connection of no-regret learning and Nash equilibrium convergence regardless of the algorithm. One way to use this result is the following: In general, it is easier to show that an algorithm is no-regret compared to showing it converges to a Nash equilibrium. Our result says that as long as we can show an algorithm is no-regret and a game satisfies a monotonicity condition, we get Nash equilibrium convergence for free. Of course, not all games satisfy the required monotonicity assumptions. But there games that do. Note that Theorem 1 is not restricted to any specific algorithm. Therefore, as we also discuss in our response to another reviewer, it is reasonable to expect that such a general result that applies to any no-regret learning algorithm will need some assumptions on the type of games it can handle that may be restrictive. Having said that, while the condition $m>2L\sqrt{N-1}$ is not necessary and, in general, can be conservative, we present a counterexample showing that the best-response algorithm may not converge to a Nash equilibrium when $m<L\sqrt{N-1}$; see Fig. 2 in our experimental results. This shows that the proposed sufficient condition $m>2L\sqrt{N-1}$ in some cases can be tight. We believe this result is interesting and supports the validity of the proposed strong monotonicity assumption.
>
> 3. Gradient descent is trivial generalization of convergence rate in strongly convex minimization: We agree that the first-order method analyzed in this paper is a trivial generalization of strongly convex minimization. However, for the zeroth-order method, we provide some new results: The smoothed game need not be strongly monotone, and Nash equilibrium convergence is still guaranteed. Moreover, we show that even if there are many equilibria of the smoothed game, they can be made close enough by appropriately designing the parameter $\delta$.
>
> 4. Best response dynamics: Although the ExtraGradient and Optimistic methods (see equations (EG) and (OG) in [2]) approximate the implicit best-response problem, we still believe our best-response algorithm is much different compared to these two methods. The most obvious difference in terms of updating actions is that these two methods which belong to the class of gradient descent methods, both need to design the step size. Instead, our best-response method in (10) does not involve the design of the step size. Since the dynamics of the EG in [2] are different from the method proposed here, the used techniques in the proofs are also different. To the best of our knowledge, there is no work analyzing the convergence of the proposed best-response method.
>
> 5. Hardness result: [3] provides a PPAD-ness completeness for strongly convex games with even 2 players. In our humble opinion, the property of strong convexity illustrates the impact of each agent's action $x_i$ on its own cost function $C_i(x_i,x_{-i})$, but has no relation to the impact of other agents' action $x_{-i}$ on the cost function $C_i(x_i,x_{-i})$. Therefore, even with strong convexity, the game is still hard to analyze because we do not know how the agents influence each other. In contrast, the strong monotonicity condition illustrates the impact of other agents' actions. We believe that, as mentioned in the Conclusion of [3], strongly monotone games may lie within the class CLS. This is an interesting topic for future work.
> 6. The usage of the word online class of games: Thanks for the suggestion, and we have made a modification.
>
>
> [1]: Golowich N, Pattathil S, Daskalakis C. Tight last-iterate convergence rates for no-regret learning in multi-player games[J]. Advances in neural information processing systems, 2020, 33: 20766-20778.
>
> [2]: Azizian W, Mitliagkas I, Lacoste-Julien S, et al. A tight and unified analysis of gradient-based methods for a whole spectrum of differentiable games[C]//International Conference on Artificial Intelligence and Statistics. PMLR, 2020: 2863-2873.
>
> [3]: Papadimitriou C H, Vlatakis-Gkaragkounis E V, Zampetakis M. The Computational Complexity of Multi-player Concave Games and Kakutani Fixed Points[J]. arXiv preprint arXiv:2207.07557, 2022.

---

> > ### Comment · Reviewer_yhnp · 2022-11-20
> > **Response about novelty**
> >
> > To clarify my intention to reject, all the papers that I referred in my initial review show their results with the following strategy:
> >
> > <<They prove that the algorithm is no-regret implying the convergence to a Nash equilibrium. >>
> >
> > Crucial Question: Please compare your strategy proof with [1] for the case of strongly monotone games.
> >
> > 1]: Golowich N, Pattathil S, Daskalakis C. Tight last-iterate convergence rates for no-regret learning in multi-player games[J]. Advances in neural information processing systems, 2020, 33: 20766-20778.

---

> > > ### Author Response · Authors · 2022-11-21
> > > **Thank you for your review. See ansewers below.**
> > >
> > > Thank you for your clarification. [1] shows that the no-regret dynamics of Optimistic gradient (OG) algorithm achieves Nash equilibrium (NE) convergence in a monotone game and provides algorithm-dependent lower bound analysis for first-order p-stationary canonical linear iterative (p-SCLI) algorithm.
> > >
> > > 1. Compared with our first result Theorem 1, the key difference is that the results in [1] are algorithm-dependent, while our Theorem 1 is algorithm-independent and not restricted to a specific class of algorithms. In the technical perspective, the proof of \textbf{all} results in [1] requires knowledge of the variable updating equations while our Theorem 1 does not require it. From this perspective, the proofs of our Theorem 1 and in [1] are certainly different.
> > > Specifically, we upper bound the Nash equilibrium errors by the Regret term (defined in equation (3)), where the Regret term is given as an oracle. The convergence analysis in [1] is directly upper bounding the Nash equilibrium errors by knowing specific type of algorithms. For example, the proof of Theorems 5 and 7 in [1] requires to know the updating equations (3) and (7), respectively. In fact, all the results in [1] cannot be obtained if the specific updating equations are unknown. Therefore, the techniques in [1] cannot be applied to the analysis of our Theorem 1 when the specific updating equations are unknown.
> > >
> > >
> > > 2. In this paper, we also explore specific algorithms (gradient descent algorithms and best-response algorithm) to relax the condition. The algorithm in [1] is in the class of first-order gradient descent and much different from our zeroth-order algorithm and the best-response algorithm. Besides, there are also some new results for the zeroth-order algorithm and the best-response algorithm, and please see Items 3 and 4 in the first author response for more details.

---

### Official Review · Reviewer_MydF · 2022-11-06

**Confidence:** 2
**Clarity, Quality, Novelty And Reproducibility:** The paper is written clearly.
**Correctness:** 4
**Technical Novelty And Significance:** 2
**Empirical Novelty And Significance:** 2
**Recommendation:** 5

**Strength And Weaknesses:**

The results are interesting but not extremely novel, given the very strong assumptions on the games that are analyzed.

**Summary Of The Paper:**

The paper shows that any no-regret dynamic converges to a Nash equilibrium in strongly convex games, assuming the convexity is sufficiently strong relative to the Lipschitz constant of the gradient function. Somewhat better bounds are shown for specific no-regret dynamics.

**Summary Of The Review:**

There seem to be few if any concrete examples where these results demonstrate something that was not previously known. The general statement is, of course, interesting. The strong assumptions reduce the attractiveness of the paper. Is it so surprising that under very strong convexity assumptions no-regret converges to Nash?

---

> ### Author Response · Authors · 2022-11-15
> **Thank you for your review. See answers below.**
>
> First note that we claim that no-regret learning converges to a Nash equilibrium in strongly monotone games rather than strongly convex games. In this paper, we attempt to answer this important question: Can no-regret learning lead to Nash equilibrium? It is well-known that the answer is in general no (see “P Mertikopoulos, Cycles in adversarial regularized learning, 2018). In this work, we present Theorem 1 that provides a positive answer for the class of $m$-strongly monotone games with $m>2L\sqrt{N-1}$, regardless of the learning algorithms. Note that Theorem 1 is not restricted to any specific algorithm. Therefore, it is reasonable to expect that such a general result that applies to any no-regret learning algorithm will need some assumptions on the type of games it can handle that may be restrictive. In a sense, we tradeoff generality in the selection of the algorithm for some restriction in the selection of the game. We believe that such an approach is quite novel compared to related literature that typically does the opposite; it restricts the selection of the algorithm and focuses on least restrictive conditions on the game. We also note that the sufficient condition $m>2L\sqrt{N-1}$ is not necessary and, in general, can be conservative. Nevertheless, at the same time, we present a counterexample showing that the best-response algorithm may not converge to a Nash equilibrium when $m<L\sqrt{N-1}$; see Fig. 2 in our experimental results. This shows that the proposed sufficient condition $m>2L\sqrt{N-1}$ in some cases can be tight. We believe this result is interesting and supports the validity of the proposed strong monotonicity assumption (recall again that Theorem 1 holds for any learning algorithm as long as it is no-regret).  We finally note that the strong monotonicity assumption on the class of games and the Lipschitz continuity assumption of the function $\nabla_i C_i(x)$ w.r.t. $x_{-i}$ needed to show Theorem 1 hold in many applications, e.g., Cournot games, retailer pricing games and Kelly auction games. Note that Theorem 1 does not need Assumptions 1 & 2. To the best our knowledge, this is the first paper to provide a connection between no-regret learning (regardless of the algorithm) and Nash equilibrium convergence.
>
> In addition to Theorem 1, we further explore first-order and zeroth-order gradient descent algorithms as well as the best-response algorithm and provide relaxed sufficient conditions for convergence to Nash equilibrium. Specifically, for the best-response algorithm and for convex games with continuous actions we show that convergence to a Nash equilibrium is guarantees if the sufficient condition $m> L\sqrt{N-1}$ is satisfied. To the best of our knowledge, similar results for the best response algorithm have not been proposed in the literature. Note that the analysis of the best-response algorithm also does not need Assumptions 1 & 2.  For the zeroth-order algorithm, we also propose new results. We show that the smoothed game does not need to be strongly monotone and Nash equilibrium convergence can still be guaranteed. Last but not least, our assumptions 1 & 2 are common in recent literature (see "Bravo Mario, Bandit learning in concave n-person games, 2018; Tianyi Lin, Optimal no-regret learning in strongly monotone games with bandit feedback, 2021") and hold for many applications, e.g., Cournot games, Retailer pricing games, Kelly auction games.

---

> > ### Comment · Reviewer_MydF · 2022-11-18
> > **reaction**
> >
> > Yes, indeed, I wrote "convex" instead of "monotone" without paying attention. I realize you're discussing strongly monotone games.
> >
> > Indeed, no-regret strategies generally do not converge to Nash. But they do converge to the set of correlated equilibria.
> >
> > I was wondering if there are any concrete example of a game where previously it was not known that at least some no-regret dynamic converges to Nash and this result shows convergence?

---

> > > ### Author Response · Authors · 2022-11-18
> > > **Thank you for your review. See answers below.**
> > >
> > > Thank you for your quick feedback. In general, it is easier to show an algorithm is no-regret compared to showing it converges to a Nash equilibrium. Our result claims that all the no-regret algorithms are able to achieve Nash equilibrium convergence in the class of m-strongly monotone games with $m>2L\sqrt{N-1}$. This positive result made the subsequent convergence analysis meaningful. For example, the best-response algorithm presented in this paper is, to the best of our knowledge, new and not known to converge to Nash equilibrium. The best-response algorithm achieves no-regret learning in some class of $m$-strongly monotone games. Then, we can directly utilize our result (Theorem 1) to obtain convergence in the class of $m$-strongly monotone games with $m>2L\sqrt{N-1}$.

---

### Decision · Program_Chairs · 2023-01-20

**Decision:**

Reject

**Justification For Why Not Higher Score:**

The paper violated formatting guidelines, and two of the results presented in the paper were already known in the literature.

**Justification For Why Not Lower Score:**

N/A

**Metareview: Summary, Strengths And Weaknesses:**

This paper treats the question of whether no-regret learning converges to equilibrium in strongly monotone games. The paper's main contributions to this question are as follows:
- **Theorem 1:** the Cesàro average of the sequence of play converges to a Nash equilibrium under no-regret learning in strongly monotone games (modulo a condition relating the strong monotonicity constant and the smoothness level of the game).
- **Theorem 2:** under (multi-agent) stochastic gradient descent, the induced sequence of play converges to Nash equilibrium at a rate of $\mathcal{O}(1/T)$ in expectation.
- **Theorem 3:** in the bandit feedback case, the induced sequence of play converges to Nash equilibrium at a rate of $\mathcal{O}(1/T^{1/3})$ in expectation.
- **Theorem 4:** exponential convergence under the best response dynamics.

Theorem 1 is presented as a convergence result, but it actually concerns the algorithm's time-averaged state, not the actual sequence of play - so it is weaker than Theorems 2-4 in this regard. The committee was not aware of a similar statement in the literature, but it was also not clear whether it is easier to show the no-regret property for a given algorithm than to show its time-averaged convergence in a strongly monotone problem. As such, the usefulness of this theorem is not clear in general.

During the discussion phase, it was also pointed out that Theorems 2 and 3 are essentially the same as the corresponding results of Bravo et al. (2018), and that said results cannot be obtained by a no-regret analysis (the proofs certainly did not use the no-regret property). In this regard, the connection between regret minimization and convergence to Nash equilibrium was left tenuous.

Finally, albeit interesting in itself, the linear rate of Theorem 4 cannot be connected to the no-regret property: the linear rate is obtained by analyzing the BR algorithm directly, and Theorem 1 is never used in the proof of Theorem 4, so the link with regret minimization is not clear here either. [It is also worth pointing out that the BR algorithm requires solving a convex problem at each iteration, so comparing oracle complexities between BR and (S)GD can be a dubious affair.]

Because of the above reasons, the conclusion of the discussion phase was that the paper does not meet the acceptance criteria for ICLR, so a decision was reached to make a "reject" recommendation to the program committee

**Procedural note:**
Independently of the above, it was pointed out that the initially submitted version violated the ICLR 2023 formatting guidelines by a significant margin. Such violations have constituted grounds for rejection in the past, and even though the authors did submit a revised version which adhered to the formatting guidelines, it is important to ensure that all papers are held to the same standard and treated uniformly. The committee decided to discuss the paper despite the above violation in order to give actionable input to the authors for a potential resubmission.

**Summary Of Ac-Reviewer Meeting:**

N/A